# Nuclear m6A reader YTHDC1 promotes muscle stem cell activation/proliferation by regulating mRNA splicing and nuclear export

**Yulong Qiao[1,2†], Qiang Sun[2,3†], Xiaona Chen[2,3], Liangqiang He[1,2], Di Wang[4], Ruibao Su[4], Yuanchao Xue[4], Hao Sun[1]\*, Huating Wang[2,3]\***

[1]Department of Chemical Pathology, Li Ka Shing Institute of Health Sciences, The Chinese University of Hong Kong, Hong Kong, China; [2]Center for Neuromusculoskeletal Restorative Medicine (CNRM), CUHK InnoHK Centres, The Chinese University of Hong Kong, Hong Kong, China; [3]Department of Orthopaedics and Traumatology, Li Ka Shing Institute of Health Sciences, The Chinese University of Hong Kong, Hong Kong, China; [4]Key Laboratory of RNA Biology, Institute of Biophysics, Chinese Academy of Sciences, Beijing, China

**\*For correspondence:**
haosun@cuhk.edu.hk (HS);
huating.wang@cuhk.edu.hk (HW)

[†]These authors contributed equally to this work

**Competing interest:** The authors declare that no competing interests exist.

**Abstract** Skeletal muscle stem cells (also known as satellite cells [SCs]) are essential for muscle regeneration and the regenerative activities of SCs are intrinsically governed by gene regulatory mechanisms, but the post-transcriptional regulation in SCs remains largely unknown. N(6)-methyladenosine (m6A) modification of RNAs is the most pervasive and highly conserved RNA modification in eukaryotic cells; it exerts powerful impact on almost all aspects of mRNA processing that is mainly endowed by its binding with m6A reader proteins. In this study, we investigate the previously uncharacterized regulatory roles of YTHDC1, an m6A reader in mouse SCs. Our results demonstrate that YTHDC1 is an essential regulator of SC activation and proliferation upon acute injury-induced muscle regeneration. The induction of YTHDC1 is indispensable for SC activation and proliferation; thus, inducible YTHDC1 depletion almost abolishes SC regenerative capacity. Mechanistically, transcriptome-wide profiling using LACE-seq in both SCs and mouse C2C12 myoblasts identifies m6A-mediated binding targets of YTHDC1. Next, splicing analysis defines splicing mRNA targets of m6A-YTHDC1. Furthermore, nuclear export analysis also leads to the identification of potential mRNA export targets of m6A-YTHDC1 in SCs and C2C12 myoblasts; interestingly, some mRNAs can be regulated at both splicing and export levels. Lastly, we map YTHDC1 interacting protein partners in myoblasts and unveil a myriad of factors governing mRNA splicing, nuclear export, and transcription, among which hnRNPG appears to be a bona fide interacting partner of YTHDC1. Altogether, our findings uncover YTHDC1 as an essential factor controlling SC regenerative ability through multifaceted gene regulatory mechanisms in mouse myoblast cells.

## Editor's evaluation

This valuable study has convincingly identified a specific regulator in skeletal muscle regeneration through a series of elegant experiments. It will form a foundation for further mechanistic investigation. The work will be of future importance in the clinical management of muscle injury and promotion of regeneration.

## Introduction

Skeletal muscle has a robust regenerative capacity, with rapid re-establishment of full power occurring even after severe damage that causes widespread myofiber necrosis. This is accomplished by satellite cells (SCs) that normally lie quiescent and uniquely labeled by the expression of paired box (Pax) transcription factor Pax7 (*Fujita and Crist, 2018*). Upon injury, the master myogenic regulator MyoD is expressed to enable rapid activation of SCs, which then expand as proliferating myoblasts; the myoblasts then differentiate and fuse into myofibers to repair the damaged muscle, while a subset self-renews to restore the quiescent SC pool. Deregulated SC activity contributes to the development of many muscle diseases; it is thus imperative to understand the way SCs contribute to regeneration. Intrinsically, both transcriptional and post-transcriptional gene regulations constitute key mechanisms governing SC activities (*Chen et al., 2021b*; *Relaix et al., 2021*). In this study, we investigate the post-transcriptional gene regulation mediated by N(6)-methyladenosine (m6A) modification.

m6A is the most pervasive and highly conserved RNA modification in eukaryotic cells and exerts powerful impact on almost all aspects of mRNA processing, including alternative splicing, nuclear export, stability maintenance, and translational efficiency, thus regulating diverse cellular processes (*Shi et al., 2019*). The past decade has witnessed rapid technical and conceptual advances that allowed transcriptome-wide interrogation of m6A dynamics and galvanized interest in m6A function. It is widely accepted (*Murakami and Jaffrey, 2022*; *Shi et al., 2019*) that RNA m6A modification is catalyzed by a multicomponent methyltransferase complex (the 'writer') comprising METTL14, METTL3, WTAP, etc., and can be removed by specific demethylases (the 'erasers'), including FTO and ALKBH5. However, it is the so-called 'reader' proteins that bind m6A and impart the epitranscriptomic information engraved in RNA m6A to functional signals; thus, these proteins are considered key effectors and executors of m6A functions. So far, a category of m6A readers has been identified and classified as several families, among which YTH domain family proteins, YTHDF1, YTHDF2, YTHDF3, YTHDC1, and YTHDC2, specifically recognize m6A sites through YTH domain and are the most well-known readers.

YTHDC1 is a unique m6A reader because of its dominant location in the nucleus endowing its post-transcriptional regulatory functions such as pre-mRNA splicing (*Kasowitz et al., 2018*; *Xiao et al., 2016*), mRNA export (*Lesbirel et al., 2018*; *Roundtree et al., 2017*), and mRNA stabilization (*Shima et al., 2017*; *Widagdo et al., 2022*; *Zhang et al., 2020*). The pioneer study from (*Xiao et al., 2016*) shows that YTHDC1 promotes exon inclusion of targeted mRNAs in Hela cells by recruiting pre-mRNA splicing factor SRSF3 while blocking SRSF10 binding (*Xiao et al., 2016*); however, in a separate study (*Roundtree et al., 2017*), YTHDC1 interacting with SRSF3 facilitates the delivery of methylated mRNAs to export receptor NXF1 for cytoplasmic export, pointing to YTHDC1 and SRSF3 as adaptor proteins coupling mRNA splicing and export (*Roundtree et al., 2017*). Interestingly, rapidly evolving evidence from the past two years implicates YTHDC1 in epigenetic and transcriptional control (*Kan et al., 2022*; *Wei and He, 2021*; *Akhtar et al., 2021*), (*Lee et al., 2021*; *Xu et al., 2022*). There is thus a current advent in illuminating molecular underpinnings of m6A-dependent functions of YTHDC1 as well as in demonstrating its biological importance in vivo. In this project, we elucidate m6A-YTHDC1 functional mechanisms in skeletal muscle stem cells and muscle regeneration.

Our findings identify YTHDC1 as an essential regulator of SC activation/proliferation in the acute injury induced muscle regeneration. Inducible YTHDC1 knockout impairs SC activation and proliferation, which blocks muscle regeneration. Mechanistically, combining transcriptome-wide YTHDC1 binding profiles and global m6A map we define m6A-YTHDC1 regulatory targets in myoblasts. Further analyses demonstrate that YTHDC1 loss substantially alters target mRNA splicing; in addition, it also impacts nuclear export of a subset of target mRNAs. Lastly, we uncover that YTHDC1 interacts with a myriad of proteins in proliferating myoblasts, including regulators of mRNA splicing, mRNA nuclear export, as well as transcriptional regulators. We further demonstrate that hnRNPG interacts with YTHDC1 and synergistically promotes myoblast proliferation. Altogether, our findings suggest that YTHDC1 is a previously uncharacterized factor governing SC activities and muscle regeneration, and it can exert pleiotropic gene regulatory functions by interacting with different proteins in myoblast cells.

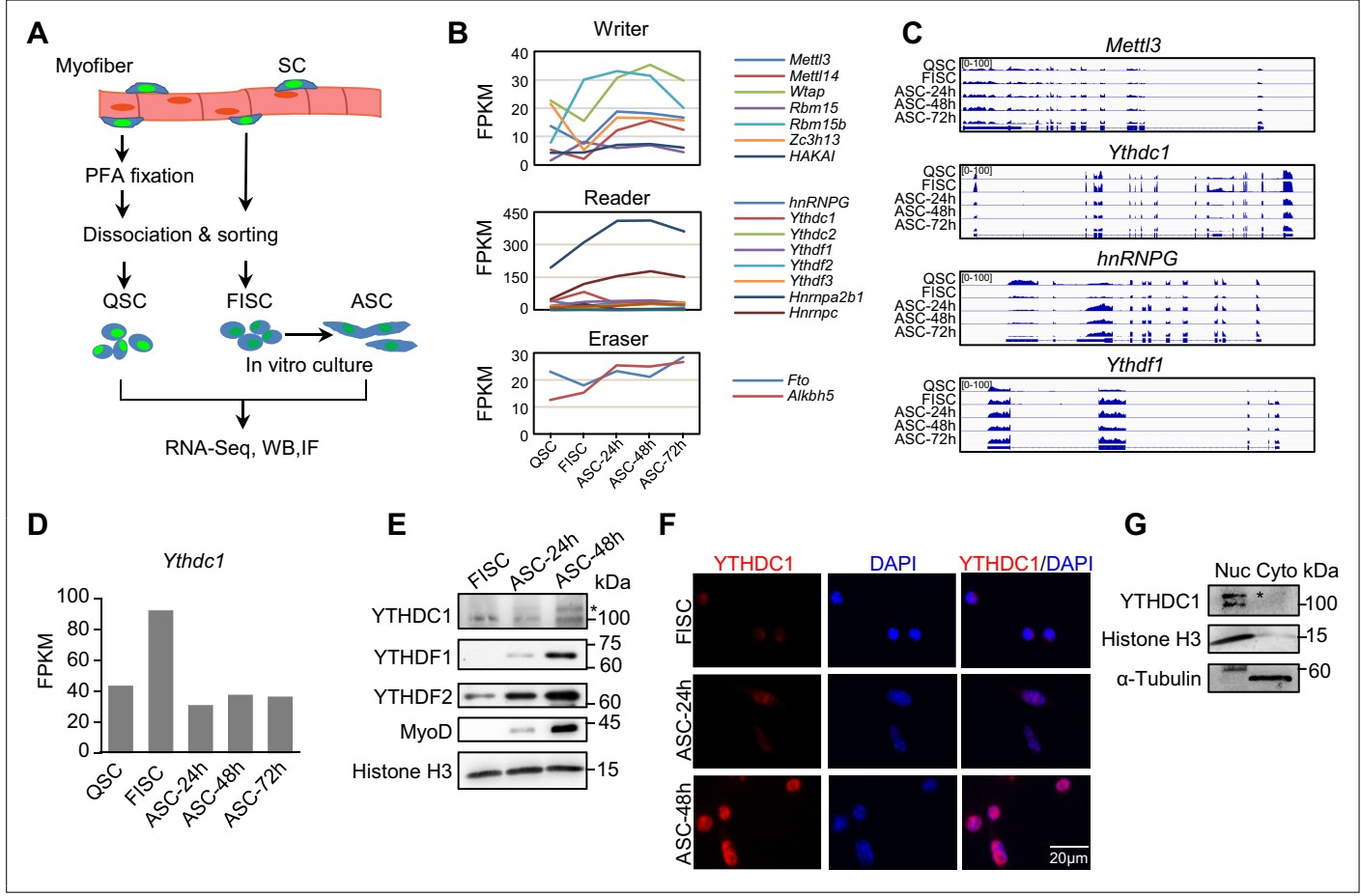

**Figure 1.** m6A regulators are dynamically expressed during satellite cell (SC) lineage progression and YTHDC1 is induced upon SC activation/proliferation. (**A**) Schematic illustration of SC collection from Pax7-nGFP mice. Fixed quiescent SCs (QSC), freshly isolated SCs (FISC), and cultured SCs (ASC) were subject to RNA-seq, western blotting (WB), and Immunofluorescence (IF) analyses. (**B**) The expression dynamics of m6A writer, reader, and eraser proteins in the above cells from analyzing the RNA-seq data. (**C**) Representative RNA-seq tracks showing the expression dynamics of the selected m6A regulators. (**D**) The expression dynamic of *Ythdc1* mRNA (FPKM) from RNA-seq. (**E**) WB showing the induction of YTHDC1, YTHDF1, and YTHDF2 proteins upon SC activation and proliferation. * denotes the correct position of YTHDC1. Histone H3 was used as a loading control. (**F**) IF staining showing the induction of YTHDC1 protein upon SC activation and proliferation. Scale bar = 20 μm. (**G**) WB showing the predominant location of YTHDC1 in nuclear portion of C2C12 myoblasts.

The online version of this article includes the following source data for figure 1:

**Source data 1.** RNA-seq measured expression.

**Source data 2.** Uncropped blot images of *Figure 1E and G*.

# Results

## m6A regulators are dynamically expressed during SC lineage progression, and YTHDC1 is induced upon SC activation/proliferation

To investigate possible roles of m6A regulators in SCs, we analyzed the expression dynamics of a panel of m6A writers, readers, and erasers using our recently generated RNA-seq datasets (*He et al., 2021*) from SCs at various time points of lineage progression (*Figure 1A*). Freshly isolated SCs (FISCs) by fluorescence-activated cell sorting (FACS) are considered as early activating cells due to the disruption of their niche by the isolation process (*Machado et al., 2017*); quiescent SCs (QSCs) were obtained through a pre-fixation step with paraformaldehyde (PFA) before the dissociation to preserve the quiescence status; FISCs were cultured for 24, 48, and 72 hr to obtain fully activated, proliferating, and differentiating SCs (ASC-24 hr, -48 hr, and -72 hr). As a result, we found that many m6A regulators were dynamically expressed in the lineage progression course (*Figure 1B–C* and *Figure 1—source*

data 1), reflecting their possible importance in controlling different phases of SC activities. Specifically, we found the reader *Ythdc1* mRNA was expressed at all stages with the highest level in FISCs (*Figure 1B–D*). By western blotting (WB) and immunofluorescence (IF) (*Figure 1E and F*), we observed a very low level of YTHDC1 protein in FISC, an evident increase in activating cells (ASC-24 hr) and continued increase in proliferating cells (ASC-48 hr); YTHDF1 and YTHDF2 reader proteins appeared to show similar expression dynamics. Consistent with prior studies (*Xiang et al., 2017*; *Xiao et al., 2016*), YTHDC1 protein was predominantly detected in the nucleus but not cytoplasm (*Figure 1F*). This was also confirmed in mouse C2C12 cell line (a commonly used surrogate for ASCs) by fractionation assay (*Figure 1G*), suggesting its possible nuclear regulatory functions in myoblast cells.

## Inducible YTHDC1 deletion in SCs abolishes acute injury-induced muscle regeneration

Despite intensive investigation of YTHDC1 regulatory mechanisms, genetic evidence remains largely lacking to support its roles in biological processes. To test its biological function in SCs, we crossed a recently available *Ythdc1* floxed allele (*Kasowitz et al., 2018*) with the *Pax7^CreER^*; *Rosa26^EYFP^* mouse (*Lepper et al., 2009*) to generate an inducible knock out (iKO) mouse to inactivate YTHDC1 specifically in SCs (*Figure 2A*). After successful removal of YTHDC1 by five consecutive plus two extra doses of tamoxifen (TMX) (*Figure 2B and C*), BaCl$_2$ was injected into the tibialis anterior (TA) muscle to induce acute damage (*Chen et al., 2021b*). In both control (Ctrl) littermate and the iKO mice, massive immune cell infiltration was observed on the first day post injury (dpi) (data not shown); at 3 dpi, SCs were rapidly activated and reached a peak of proliferation in Ctrl (data not shown); by 5 dpi, they were mainly differentiating and labeled with eMyHC+ (expressed only in newly regenerating fibers) (*Figure 2D and E*), coinciding with the initiation of myofiber repair. As a striking contrast, muscle regeneration was nearly completely abolished in the iKO muscles; excessive immune infiltration was still present at 5 or 7 dpi with no signs of repair (*Figure 2D*); a sharp decrease of eMyHC+ myofibers was detected at both time points (*Figure 2E*). A complete loss of Pax7+ cells (*Figure 2F and G*) was also observed in iKO vs. Ctrl despite no difference was observed on uninjured muscles (*Figure 2F and G*). At 14 dpi, injured Ctrl muscles were largely repaired (*Figure 2—figure supplement 1A–C*). But the regeneration was never observed even at 28 dpi; iKO muscle remained significantly smaller than Ctrl (*Figure 2—figure supplement 1C*). Altogether, the above data indicate the essential function of YTHDC1 in acute injury-induced muscle regeneration.

## Inducible YTHDC1 knockout impairs SC activation/proliferation

To pinpoint the major defects of iKO SCs that cause the blocked regeneration, we suspected YTHDC1 loss impaired SC activation and proliferation considering it was induced upon SC activation and highly expressed in proliferating myoblasts (*Figure 1D and E*). To this end, FISCs from Ctrl or iKO mice were cultured for 24 or 48 hr; and indeed the iKO cells displayed evident growth arrest (*Figure 3—figure supplement 1A and B*). This was confirmed by 4 hr EdU treatment and staining; activation (24 hr) and proliferation (48 hr) were drastically inhibited in the iKO compared to the Ctrl (*Figure 3A*). On isolated single myofibers, SCs also failed to proliferate, confirmed by EdU staining of the fibers cultured for 48 hr (*Figure 3B*). Consistently, staining for Pax7 and MyoD showed that a lower percentage of double positive cells were detected at both ASC-24 hr and 48 hr in the iKO compared to the Ctrl (*Figure 3C*); the protein levels of Pax7 and MyoD were also largely diminished in the iKO cells at 48 hr (*Figure 3—figure supplement 1C*). On single myofibers, the number of Pax7+ MyoD+ SCs was also reduced at both 24 hr and 48 hr (*Figure 3D*). Lastly, in vivo EdU assay was performed (*Figure 3E*); EdU was injected 2 days after BaCl$_2$ -induced injury and SCs were isolated 12 hr later for staining. A significant reduction of EdU+ cells was observed in the iKO compared to the Ctrl muscles (*Figure 3F*). Altogether, the above findings demonstrate that YTHDC1 loss causes a severe defect in SC activation and proliferation, thus solidifying the essential function of YTHDC1 in these stages of SC activities.

Next, transcriptomic analysis using RNA-seq was conducted in ASC-24 hr and ASC-48 hr cells to validate the above observed phenotypical defects in iKO cells. As a result, a total of 547 differentially expressed genes (DEGs) were identified in iKO vs. Ctrl SCs with 431 down- and 116 upregulated in ASC-24 hr (*Figure 3G* and *Supplementary file 1*); and the downregulated genes were enriched for GO terms such as 'Nuclear division,' 'Chromosome segregation,' etc. (*Figure 3G* and *Supplementary file 1*). Similarly, a total of 1272 DEGs (556 down- and 716 upregulated) were identified in ASC-48

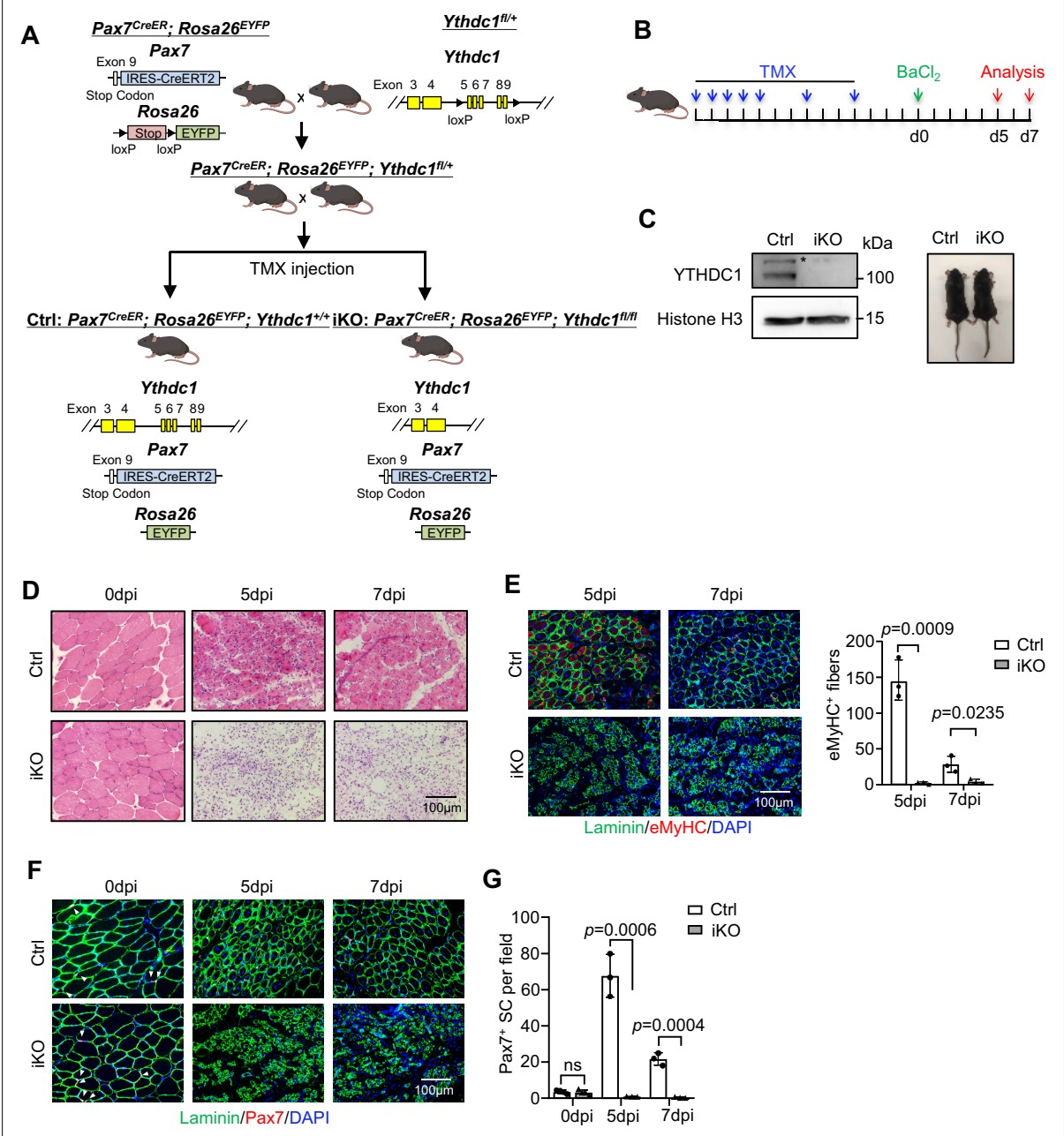

**Figure 2.** Inducible YTHDC1 deletion in satellite cells (SCs) abolishes acute injury-induced muscle regeneration. (**A**) Breeding scheme for generating YTHDC1-inducible knockout (iKO) and control (Ctrl) mice. (**B**) Schematic outline of the tamoxifen (TMX) administration used in the study and experimental design for testing the effect of YTHDC1 deletion on barium chloride ($BaCl_2$)-induced muscle regeneration process. (**C**) Left: western blotting (WB) showing the deletion of YTHDC1 in ASC-48 hr from iKO but not Ctrl mice. Right: no obvious morphological difference was detected in iKO vs. Ctrl mice. (**D**) H&E staining of the above injured muscles at 0, 5, and 7 days post injury (dpi). Scale bar = 100 µm. (**E**) Left: immunostaining of eMyHC (red) and laminin (green) of the above injured tibialis anterior (TA) muscles at 5 and 7 dpi. Scale bar = 100 µm. Right: quantification of eMyHC-positive fibers per field. n = 3 mice per group. (**F**) Immunostaining of Pax7 (red) and laminin (green) on TA muscle sections at 0, 5, and 7 dpi. Scale bar = 100 µm. (**G**) Quantification of Pax7-positive SCs per field at 0, 5, and 7 dpi. n = 4 mice per group for 0 dpi, n = 3 mice per group for 5 and 7 dpi. Bars represent mean ± SD for all graphs. Statistical significance was determined using a two-tailed Student's *t*-test.

The online version of this article includes the following source data and figure supplement(s) for figure 2:

**Source data 1.** Uncropped blot images of *Figure 2C*.

**Figure supplement 1.** Analyses of regeneration at 14 and 28 days post injury.

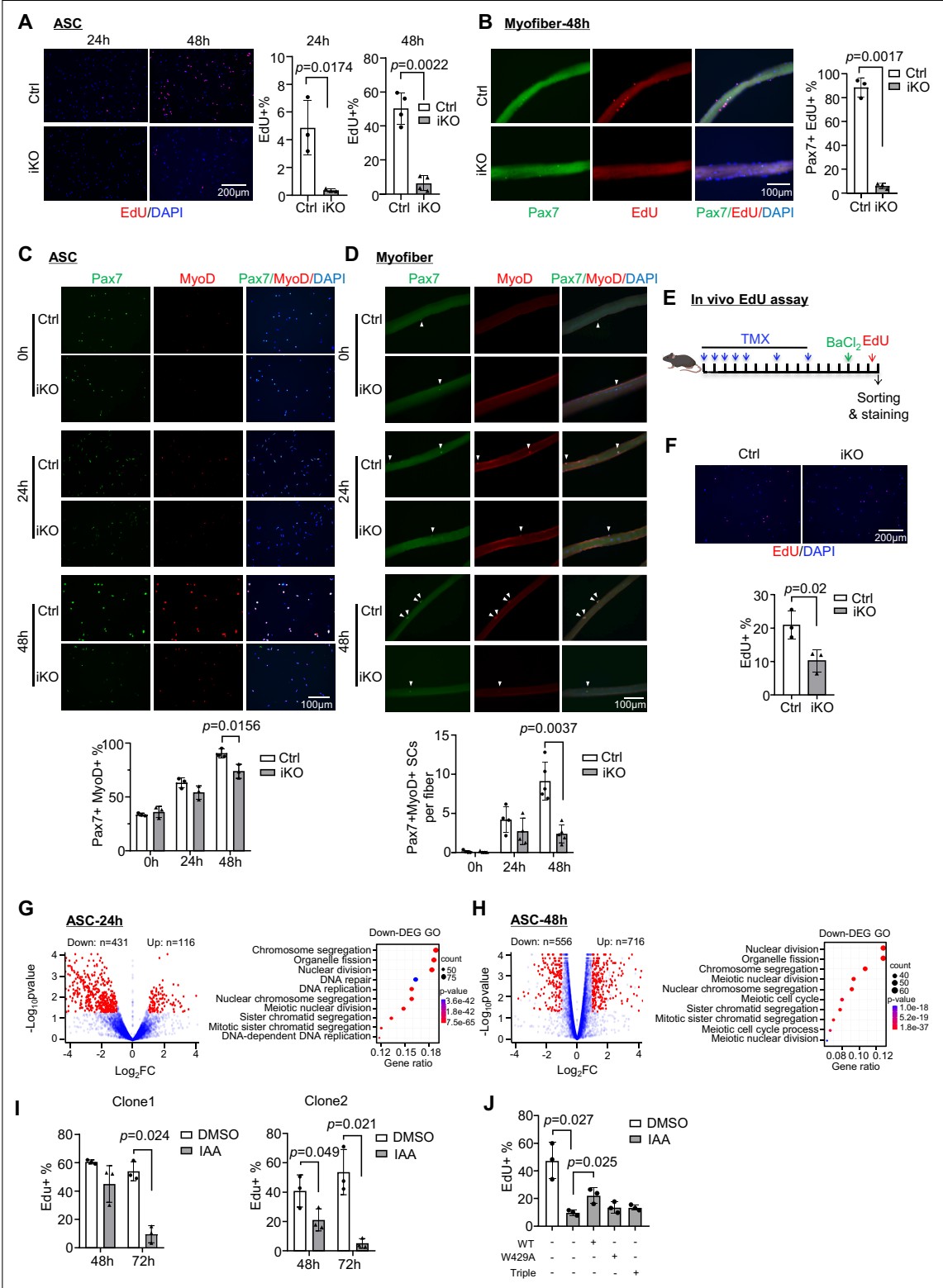

**Figure 3.** Inducible YTHDC1 knockout impairs satellite cell (SC) activation/proliferation. (**A**) Left: EdU (red) staining of ASC-24 hr and ASC-48 hr from inducible knock out (iKO) and Ctrl mice. Scale bar = 200 μm. Right: quantification of the percentage of EdU+ cells. n = 3 mice per group for ASC-24 hr, n = 4 mice per group for ASC-48 hr. (**B**) Left: EdU (red) staining of EDL myofibers isolated from Ctrl or iKO mice and cultured for 48 hr. Scale bar = 100 μm. Right: quantification of the percentage of Pax7+ EdU+ SCs. n = 3 mice per group. (**C**) Top: immunofluorescence (IF) staining of Pax7 (green) and MyoD (red) of freshly isolated SCs (FISC) (0 hr), ASC-24 hr and ASC-48 hr from Ctrl and iKO mice. Scale bar = 100 μm. Bottom: quantification of the percentage

*Figure 3 continued on next page*

*Figure 3 continued*

of Pax7+ MyoD+ cells. n = 3 mice per group. (**D**) Top: IF staining of Pax7 (green) and MyoD (red) on EDL myofibers at 0 hr (freshly isolated), 24 hr, and 48 hr. Scale bar = 100 μm. Bottom: quantification of the number of Pax7+ MyoD+ cells per fiber (n = 4 mice per group for 0 and 24 hr, n = 5 mice per group for 48 hr). (**E**) Schematic illustration of in vivo EdU assay in Ctrl and iKO mice. (**F**) Top: EdU staining of the above freshly isolated and fixed SCs at 3 dpi. Scale bar = 200 μm. Bottom: quantification of the percentage of EdU+ cells in iKO vs. Ctrl. n = 3 mice per group. (**G, H**) Left: RNA-seq was performed in ASC-24 hr or -48 hr from iKO and Ctrl mice. Volcano plot showing the down- and upregulated genes in iKO vs. Ctrl. Right: GO analysis for the downregulated genes. (**I**) Two independent mAID-YTHDC1 cell lines were treated with DMSO or 5-Ph-IAA (IAA) for the indicated time. EdU assay was performed and the percentage of EdU+ cells was quantified at the designated time points. n = 3 replicates. (**J**) WT-YTHDC1, or a single (W429A), or a triple (K362A, S363A, and N364A) mutant of YTHDC1 lacking m6A binding ability was transfected with the mAID-YTHDC1 cells then treated with IAA. mAID-YTHDC1 cells treated with DMSO or IAA were used as control. EdU assay was then performed and the percentage of EdU+ cells was quantified. n = 3 replicates. Bars represent mean ± SD for all graphs. Statistical significance was determined using a two-tailed Student's *t*-test.

The online version of this article includes the following source data and figure supplement(s) for figure 3:

**Figure supplement 1.** Analyses of activation/proliferation defect of satellite cells (SCs) and C2C12 myoblasts upon YTHDC1 loss.

**Figure supplement 1—source data 1.** Uncropped blot images of *Figure 3—figure supplement 1C,E, and I*.

hr (*Figure 3H*) and again the downregulated genes were enriched for similar GO terms as the above (*Figure 3H* and *Supplementary file 1*). These findings thus substantiate the activation/proliferation defects observed in the iKO cells.

Lastly, to further solidify the function of YTHDC1 in proliferating myoblasts, we generated a C2C12 mouse myoblast cell line with inducible YTHDC1 degradation using the auxin-inducible degron (AID2) system (*Yesbolatova et al., 2020*; *Figure 3—figure supplement 1D*). An mAID-mCherry tag was knocked into the *Ythdc1* N-terminal locus in C2C12 expressing auxin receptor Ostir1 (F74G) (*Figure 3—figure supplement 1D*). As expected, the addition of 5-Ph-IAA (auxin, IAA) but not DMSO (negative control) induced rapid degradation of the mAID-tagged YTHDC1 (*Figure 3—figure supplement 1E and F*). In line with the results obtained from ASCs, decreased proliferation rate was observed in two independent AID-YTHDC1 clones treated with IAA for both 48 hr and 72 hr (*Figure 3—figure supplement 1G and H* and *Figure 3I*), strengthening that YTDHC1 functions to promote myoblast proliferation. Furthermore, to validate the causative role of YTHDC1 in driving the above phenotype, rescue experiments were performed by overexpressing a WT or a single (W429A) or triple (K362A, S363A, and N364A) mutant to disable m6A binding of YTHDC1 (*Liu et al., 2021*; *Xu et al., 2014*) in the AID-YTHDC1 C2C12 cells. As expected, only the WT but not mutant YTHDC1 could partially rescue the proliferation defect caused by YTHDC1 degradation (*Figure 3J*, *Figure 3—figure supplement 1I*), suggesting that indeed YTHDC1 is essential to promote myoblast proliferation and this function is dependent on its m6A binding function.

## LACE-seq defines transcriptome-wide YTHDC1 binding profiles in myoblasts

To fathom the underlying mechanism, we conducted transcriptome-wide binding analysis for YTHDC1, reasoning ultimately it is the binding sites/targets that determine its function. To this end, we harnessed the recently developed LACE-seq (the Linear Amplification of Complementary DNA Ends and Sequencing) method (*Figure 4A*) that enables global profiling of RNA-binding protein (RBP) target sites with a relatively low quantity of starting cellular material (*Su et al., 2021*). Around 1 million ASC-48 hr were subject to LACE-seq with two biological replicates (B1, B2) and four technical replicates (T1, T2) (*Figure 4B*). As a result, a total of 2444 shared peaks corresponding to 951 genes were identified from comparing the four replicates and defined as YTHDC1 binding targets (*Figure 4B* and *Supplementary file 2*). All of these replicates were well correlated with each other (Pearson correlation coefficient: 0.82~0.91). These genes were enriched for GO terms such as 'Cell projection organization,' 'Cell junction assembly,' etc. (*Figure 4C* and *Supplementary file 2*). To reinforce the result, we also performed LACE-seq with C2C12 myoblasts; two technical replicates were included, and a much higher number of cells (10 million) were used for each replicate (*Figure 4A and B*). Our results showed that data from C2C12 was of very high quality with the two replicates displaying high correlation (Pearson correlation coefficient: 0.96). A total of 29107 shared peaks corresponding to 5897 genes were identified as YTHDC1 binding targets (*Figure 4B* and *Supplementary file 2*). Very similar to the above result from ASCs, these genes were also enriched for a variety of GO terms such

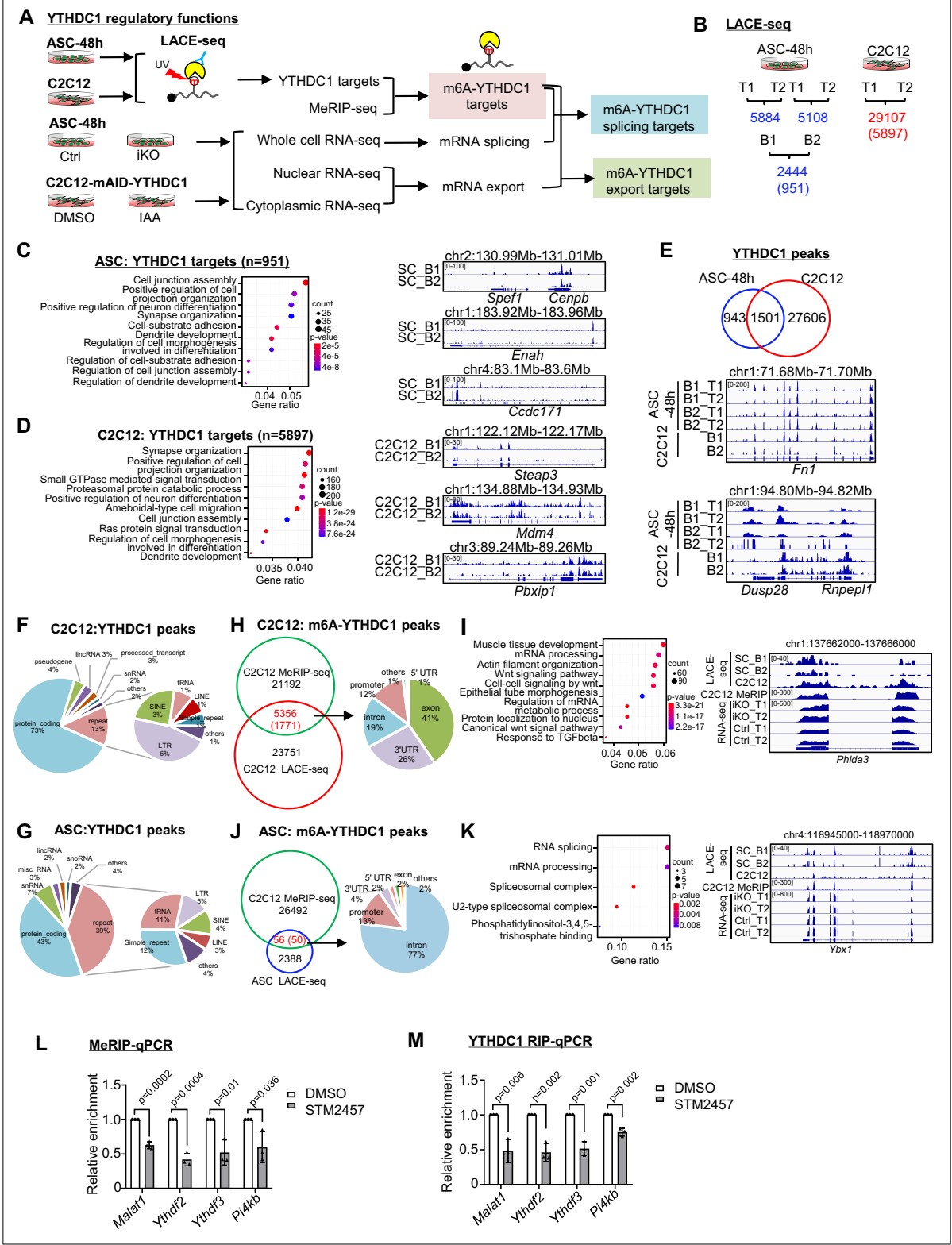

**Figure 4.** LACE-seq defines transcriptome-wide YTHDC1 binding profiles in myoblasts. (**A**) Schematic illustration of the experimental design for performing LACE-seq and subsequent combination with MeRIP-seq, bulk RNA-seq, subcellular RNA-seq for defining and elucidating YTHDC1 splicing/ export targets and post-transcriptional regulation. (**B**) LACE-seq was performed in both ASC-48 hr and C2C12 myoblasts and the number of identified peaks in each technical (T) or biological replicate (B) and the shared number of peaks (genes) between the replicates are shown. (**C**) Left: Go analysis for the identified YTHDC1 targets in ASCs. Right: genome tracks for three selected genes. (**D**) Left: Go analysis for the identified YTHDC1 targets in C2C12.

*Figure 4 continued on next page*

*Figure 4 continued*

Right: genome tracks for three selected genes. (**E**) Top: overlapping between the above-identified C2C12 and ASC peaks. Bottom: genome tracks for two selected genes. (**F, G**) Left: the genome distribution of YTHDC1 binding peaks in C2C12 or ASC. Right: detailed distribution of the YTHDC1 binding peaks on repeat regions. (**H**) Left: integrating C2C12 MeRIP-seq data with the C2C12 LACE-seq identified 5356 regions (1771 mRNAs, as m6A-YTHDC1 targets). Right: distribution of YTHDC1 binding on the above targets. (**I**) Left: GO analysis of the above 1771 targets. Right: genomic tracks of a selected target, *Phlda3*. (**J**) Left: integrating the C2C12 MeRIP-seq data with the ASC LACE-seq identified 56 regions (50 mRNAs, as m6A-YTHDC1 targets). Right: distribution of YTHDC1 binding on the above-identified target mRNAs. (**K**) Left: GO analysis of the above 50 targets. Right: genomic tracks of a selected target, *Ybx1*. (**L**) C2C12 cells were treated with DMSO or STM2457. m6A modification levels on m6A-YTHDC1 targets were examined by MeRIP-qPCR. Relative enrichment was calculated by comparing the MeRIP/input ratio of DMSO and STM2457. n = 3 biological replicates. (**M**) C2C12 cells were transfected with pRK5-flag-YTHDC1 and then treated with DMSO or STM2457. YTHDC1 binding to m6A-YTHDC1 targets was examined by RIP-qPCR. Relative enrichment was calculated by comparing the RIP/input ratio of DMSO and STM2457. n = 3 biological replicates. Bars represent mean ± SD for all graphs. Statistical significance was determined using a two-tailed Student's *t*-test.

The online version of this article includes the following figure supplement(s) for figure 4:

**Figure supplement 1.** Genome tracks of selected m6A-YTHDC1 targets.

as 'Cell projection organization,' 'Cell junction assembly,' etc. (*Figure 4D* and *Supplementary file 2*). In fact, 1501 out of the 2444 identified peaks in ASCs could be found in C2C12 (*Figure 4E* and *Supplementary file 2*), testifying the success of the assay. Next, further probing into the binding locations, we found that in both C2C12 (*Figure 4F* and *Supplementary file 2*) and ASCs (*Figure 4G* and *Supplementary file 2*), a large portion of the peaks (73% and 43%) were mapped to protein coding genes, which is largely consistent with prior reports from conducting YTHDC1 global profiling through CLIP-seq or RIP-seq in other cells (*Chen et al., 2021a*; *Liu et al., 2021*; *Patil et al., 2016*; *Xiao et al., 2016*), suggesting that regulating mRNA processing is probably the dominant role of YTHDC1 in mouse myoblasts. Interestingly, small portions of the peaks were mapped in lincRNAs, snRNAs, snoRNAs, etc., and a substantial portion was mapped to repeat regions such as SINE (4%) and LINE (3%) (*Figure 4G*), which was consistent with prior reports (*Liu et al., 2021*).

To further tease out m6A-dependent function of YTHDC1, we obtained the published methylated RNA immunoprecipitation sequencing (MeRIP-seq) data from C2C12 (*Gheller et al., 2020*), which was of good quality shown by the enrichment of a classical RRACH motif in the identified binding sites and also dominant enrichment found at 3'UTR genomic regions (data not shown). By integrating the dataset with the above C2C12 LACE-seq dataset, a total of 5356 peaks (1771 genes) were identified; 41% of the peaks resided in exons followed by 3'UTRS (26%), introns (19%), and promoters (12%) (*Figure 4H* and *Supplementary file 2*). The above-defined final set of 1771 m6A-YTHDC1 targets were enriched for variable GO terms (*Figure 4I* and *Supplementary file 2*), among which Wnt signaling pathway is related to myoblast proliferation (*Figure 4I* and *Supplementary file 2*; *Otto et al., 2008*). Besides, a large group of m6A-YTHDC1 targets was enriched for mRNA processing (*Figure 4I*). Interestingly, *Ythdf2* and *Ythdf3* were found on the list (*Figure 4—figure supplement 1* and *Supplementary file 2*). Next, we also intercepted the ASC LACE-seq with the above C2C12 MeRIP-seq data and identified a total of 56 peaks (corresponding to 50 genes) (*Figure 4J* and *Supplementary file 2*); interestingly, these genes were highly enriched for GO terms such as 'RNA splicing,' 'mRNA processing,' etc. (*Figure 4K* and *Supplementary file 2*), which were also found in C2C12 (*Figure 4I*), suggesting that m6A-YTHDC1 not only modulates RNA processing but also controls the processing of RNA processing factors. To solidify the findings that YTHDC1 regulates its targets through m6A binding, we inactivated m6A writer METTL3 using its inhibitor STM2457 (*Yankova et al., 2021*) in C2C12 and examined the binding of YTHDC1 to three selected m6A-YTHDC1 targets, *Ythdf2*, *Ythdf3*, *Pi4kb*, using *Malat1* as a positive control (*Wang et al., 2021*). As expected, STM2457 treatment significantly reduced the m6A modification level on the three targets as revealed by meRIP-qPCR (*Figure 4L*); concomitantly, YTHDC1 binding assessed by RIP-qPCR was also decreased (*Figure 4M*), indicating indeed the regulatory function of YTHDC1 relies on m6A binding.

## YTHDC1 depletion in ASCs leads to altered splicing events

Next we tested whether YTHDC1 plays a role in modulating mRNA splicing that represents its best known function (*Kasowitz et al., 2018*; *Xiao et al., 2016*). To this end, we analyzed the splicing events using the RNA-seq data from *Figure 3G and H* (*Figure 5A*). Expectedly, in both ASC-24 hr

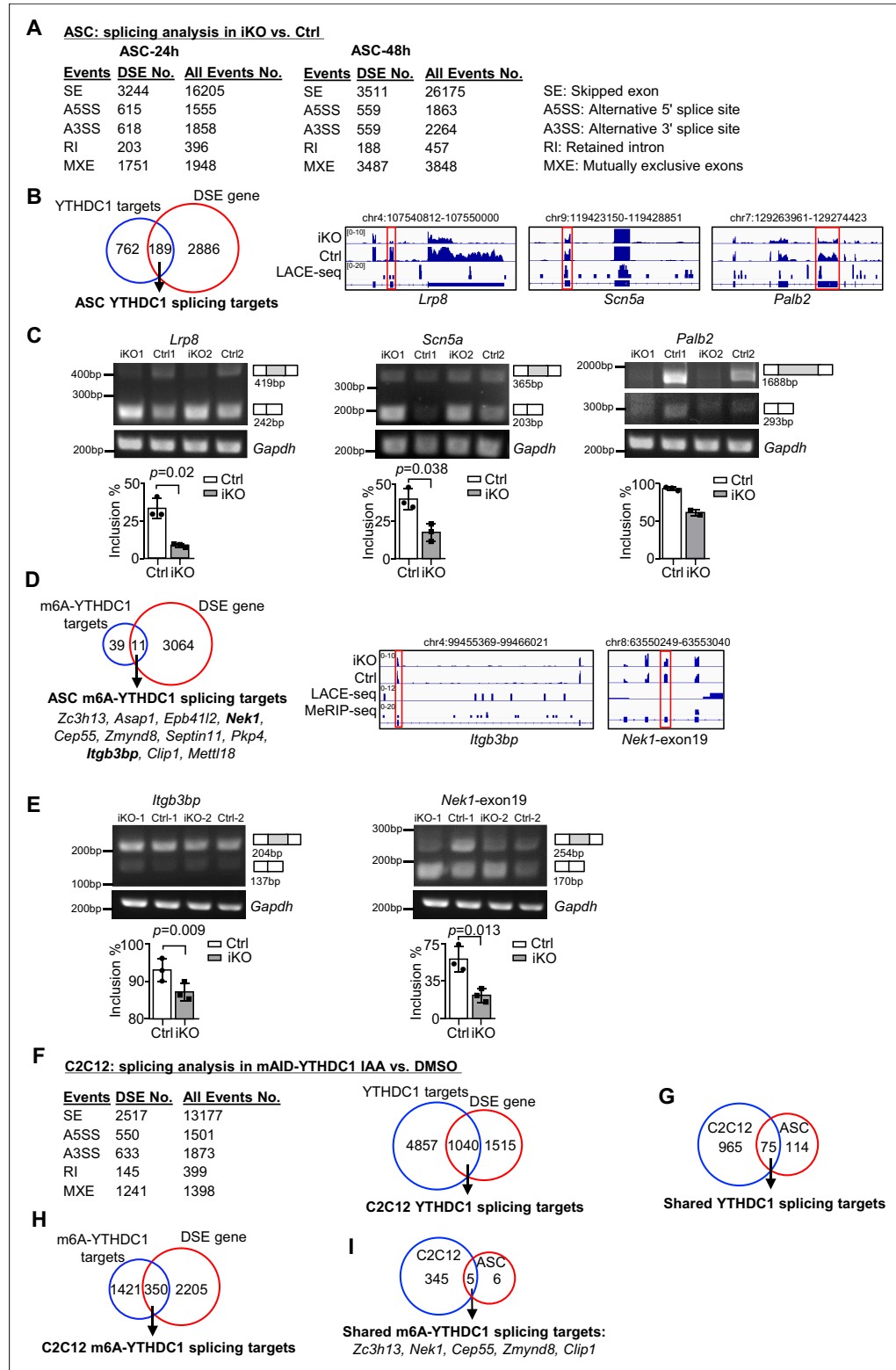

**Figure 5.** YTHDC1 depletion in ASCs leads to altered splicing events. (**A**) Splicing analysis based on bulk RNA-seq data from ASC-24 hr or -48 hr defined five types of splicing events. The total number of each event in Ctrl, and the differential spliced events (DSE) in inducible knock out (iKO) vs. Ctrl are shown. (**B**) Left: combining the above ASC-48 hr DSEs with the ASC LACE-seq targets identified a total of 189 YTHDC1 splicing target mRNAs. Right: genome tracks of three selected targets. (**C**) Top: RT-PCR assay was performed in ASC-48 hr from YTHDC1-iKO and Ctrl to

*Figure 5 continued*

verify altered splicing of the three selected target mRNAs, *Palb2*, *Lrp8*, and *Scn5a*. *Gapdh* was used as a control. Bottom: quantification of exon inclusion level. Exon inclusion level was defined as the percentage of transcripts that includes the specific exon. Included/(included + skipped). n = 3 mice per group for *Lrp8* and *Scn5a*. (**D**) Left: combining the above ASC-48 hr DSE with m6A-YTHDC1 targets uncovered 11 m6A-YTHDC1 splicing targets. Right: genome tracks of two selected targets. (**E**) Top: RT-PCR assay was performed in ASC-48 from YTHDC1-iKO and Ctrl to verify altered splicing of the two selected target mRNA, *Itgb3bp*, and *Nek1*. *Gapdh* was used as a control. Bottom: quantification of exon inclusion level. n = 3 mice per group. (**F**) Left: splicing analysis based on bulk RNA-seq data from C2C12-mAID-YTHDC1 cells with or without YTHDC1 degradation. Right: combining the above C2C12 DSEs with the C2C12 LACE-seq targets identified a total of 1040 YTHDC1 splicing target mRNAs. (**G**) Overlapping between the above-identified YTHDC1 splicing targets in ASC-48 hr and C2C12. (**H**) Combining the above C2C12 DSEs with m6A-YTHDC1 targets uncovered 350 m6A-YTHDC1 splicing targets. (**I**) Overlapping of m6A-YTHDC1 splicing targets in C2C12 and ASC-48 hr. Bars represent mean ± SD for all graphs. Statistical significance was determined using a two-tailed Student's *t*-test.

The online version of this article includes the following source data and figure supplement(s) for figure 5:

**Source data 1.** Uncropped gel images of *Figure 5C*.

**Source data 2.** Uncropped gel images of *Figure 5E*.

**Figure supplement 1.** GO analysis of YTHDC1 splicing targets in ASCs and C2C12.

and -48 hr, a high number of differential splicing events (DSEs) were uncovered in the iKO compared to the Ctrl cells (*Figure 5A*); skipped exons (SEs) and mutually exclusive exons (MXEs) dominated these events, which was consistent with previously reported role for YTHDC1 in promoting exon inclusion (*Xiao et al., 2016*). Among the 951 targets of YTHDC1, a total of 189 displayed DSE (*Figure 5B* and *Supplementary file 3*), thus defined as YTHDC1 splicing targets. These targets were enriched for 'Regulation of actin filament-based process' and 'Dendrite development' (*Figure 5—figure supplement 1A* and *Supplementary file 3*), among which we experimentally validated the altered splicing on *Palb2*, *Scn5a*, *Lrp8* mRNAs by RT-PCR assay (*Figure 5B and C*). To pinpoint m6A-YTHDC1-dependent splicing events, we then intercepted the DSEs with the 50 m6A-YTHDC1 targets identified in ASCs (*Figure 4J*). As a result, 11 of them were defined as m6A-YTHDC1 splicing targets (*Figure 5D*), among which the altered splicing on *Itbp3bp* and *Nek1* mRNAs was experimentally validated (*Figure 5D and E*).

To ascertain the above findings, we then performed RNA-seq in C2C12 cells with inducible YTHDC1 deletion (cells were treated with IAA for 8 hr to capture the immediate effect only). A total of 2555 genes with DSE were detected upon YTHDC1 degradation and a large portion (1040) were its binding targets defined in *Figure 4D* (*Figure 5F* and *Supplementary file 3*). This evidence suggests that YTHDC1 possibly plays a dominant role as a splicing regulator in C2C12. Of note, these targets were highly enriched for 'Chromatin organization' and 'Histone modification' (*Figure 5—figure supplement 1B* and *Supplementary file 3*). Also, 75 of these YTHDC1 splicing targets were found in ASC (*Figure 5G*) and enriched for 'Transcription coregulator activity' and 'Transcription corepressor activity' (*Figure 5—figure supplement 1C* and *Supplementary file 3*). Furthermore, 350 out of the 1771 m6A-YTHDC1 targets identified in C2C12 cells (*Figure 4H*) were differentially spliced upon YTHDC1 degradation (*Figure 5H*) and enriched for 'mRNA processing,' 'regulation of mRNA metabolic process,' 'Wnt signaling pathway,' etc. (*Figure 5—figure supplement 1D* and *Supplementary file 3*); and 5 of these targets were shared in ASCs, including *Zc3h13*, *Nek1*, *Cep55*, *Zmynd6*, and *Clip1* mRNAs (*Figure 5I*). Of note, NEK1 is known to play a role in regulating cell cycle (*Chen et al., 2008*). Altogether, these findings lead us to conclude that m6A-YTHDC1 orchestrates splicing of mRNAs that have important functions in myoblast cells, and its role in promoting myoblast proliferation is mediated through a wide spectrum of mRNA targets.

## YTHDC1 loss inhibits mRNA nuclear export

In addition to splicing regulation, we also wondered whether YTHDC1 plays a role in controlling mRNA nuclear export as hinted by prior studies (*Lesbirel et al., 2018*; *Roundtree et al., 2017*). To test this notion, nuclear (nuc) and cytoplasmic (cyto) fractions were prepared from ASC-48 hr cells of Ctrl or YTHDC1 iKO muscles and subject to RNA-seq, respectively. As a result, a total of 11,637

mRNAs were found to be expressed in cyto or nuc fractions and the cyto/nuc ratio was calculated to assess the change of mRNA distribution after YTHDC1 knock out. No significant difference was detected in the average cyto/nuc ratio of these expressed mRNAs in iKO vs. Ctrl (*Figure 6—figure supplement 1A and B* and *Supplementary file 4*), suggesting that YTHDC1 loss may not cause global alteration in mRNA nuclear export. Nevertheless, 1045 mRNAs did show altered cyto/nuc ratio and they were enriched for 'Methylation,' 'Chromatin modification,' etc. (*Figure 6—figure supplement 1C* and *Supplementary file 4*). To specifically examine whether YTHDC1 binding could directly modulate nuclear export, we found a total of 687 of the above-identified YTHDC1 binding targets were expressed in cyto or nuc fractions; no significant difference was detected in the average cyto/nuc ratio of these mRNAs in the iKO compared to the Ctrl (*Figure 6A and B* and *Supplementary file 4*), suggesting that YTHDC1 binding may not have a major impact on global nuclear export of its target mRNA. Nevertheless, 54 mRNAs did display significant cyto/nuc decrease in iKO vs. Ctrl thus defined as YTHDC1 export targets (*Figure 6C and D* and *Supplementary file 4*), and they were enriched for 'Protein serine kinase activity' ( *Figure 6—figure supplement 1G* and *Supplementary file 4*). Of note, 13 were also defined as YTHDC1 splicing targets (*Figure 6C*), suggesting that YTHDC1 may simultaneously regulate both splicing and export of these target mRNAs.

To solidify YTHDC1 regulating myoblast mRNA nuclear export, we performed the above assay/analysis in C2C12 mAID-YTHDC1 cells. Similarly, no global impact on the distribution of all expressed mRNAs (11175) was detected upon YTHDC1 degradation (*Figure 6—figure supplement 1D and E* and *Supplementary file 4*). When analyzing YTHDC1 bound mRNAs, a much higher number (5000) compared to the ASC (because of the higher number of LACE-seq targets from C2C12; *Figure 4*) was found to be expressed in cyto or nuc fractions and a significant decrease in the average cyto/nuc ratio was found upon YTHDC1 degradation (*Figure 6E and F* and *Supplementary file 4*), suggesting that YTHDC1 loss caused global impact on its target mRNA nuclear export in C2C12. Similarly, a total of 813 binding targets were defined as YTHDC1 export targets and displayed reduction in cyto/nuc ratio upon YTHDC1 degradation (*Figure 6G–I* and *Supplementary file 4*). These targets were enriched for 'Histone modification,' 'Chromatin organization,' etc. (*Figure 6G* and *Supplementary file 4*). About 25% of them were also identified as YTHDC1 splicing targets (*Figure 6H*). Similarly, a total of 322 m6A-YTHDC1 export targets were defined in C2C12 cells and also enriched for 'Histone modification,' 'Chromatin organization,' etc. (*Figure 6J–N* and *Supplementary file 4*). And 75 of them were also splicing targets of C2C12 m6A-YTHDC1 (*Figure 6M* and *Supplementary file 4*). Altogether, the above results demonstrate that YTHDC1 binding can indeed promote nuclear export of target mRNAs in myoblast cells, suggesting that YTHDC1 plays pleiotropic roles in regulating both splicing and export of mRNAs in myoblasts.

## Co-IP/MS leads to the identification of YTHDC1 interacting partners

Lastly, to further fathom the mechanism underlying the above-identified YTHDC1 regulatory functions in myoblasts, we performed co-immunoprecipitation (co-IP) coupled with mass spectrometry (MS) to identify its protein interactome knowing that the function of YTHDC1 is largely mediated by its transcriptionally or post-transcriptionally interacting partners (*Widagdo et al., 2022*). To this end, Flag-tagged YTHDC1 or a pRK5-vector plasmid was transfected in C2C12 myoblasts (*Figure 7A and B*); the interacting proteins were retrieved by anti-Flag beads and subject to MS analysis (*Figure 7A*). The results identified a total of 912 potential interacting partners (*Figure 7C* and *Supplementary file 5*). Of note, they were highly enriched for RNA splicing and mRNA export factors (*Figure 7D*), which was in accordance with the above findings, solidifying that the dominant functions of YTHDC1 in myoblasts are splicing and export regulations.

Among the splicing factors (*Figure 7E*), hnRNPG (also called RNA binding motif protein X [RBMX]) ranked very high (no. 14) on the list. It is a ubiquitously expressed RBP best known for its function in splicing control (*Elliott et al., 2019*; *Heinrich et al., 2009*). Their physical interaction was validated by performing co-IP followed by WB using both exogenously (*Figure 7H*) and endogenously (*Figure 7I*) expressed YTHDC1 in C2C12 myoblasts; additionally, the interaction was also validated in 293T cells (*Figure 7J*). To demonstrate the functional relevancy of hnRNPG in SCs, we knocked down hnRNPG in ASCs and found cell proliferation was reduced (*Figure 7K*), which phenocopied YTHDC1 loss caused defect. The finding thus demonstrates that hnRNPG functions synergistically with YTHDC1 in promoting SC proliferation. Furthermore, we also identified the well-known binding

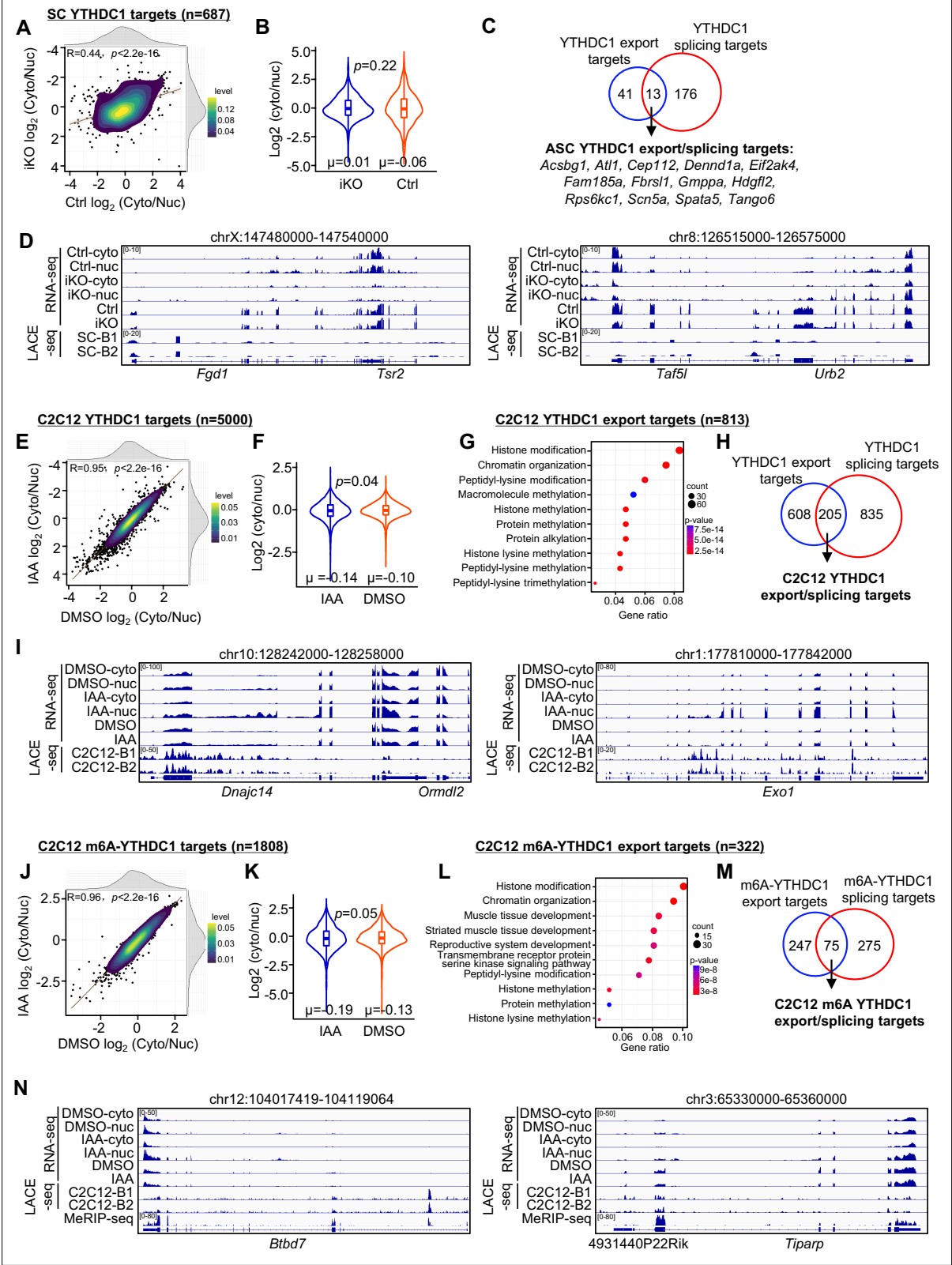

**Figure 6.** YTHDC1 loss inhibits mRNA nuclear export. (**A**) Subcellular RNA-seq was performed using cytoplasmic and nuclear fractions isolated from ASC-48 hr of Ctrl and YTHDC1 inducible knock out (iKO) mice. The $\log_2$(cyto/nuc) expression change was calculated for YTHDC1 targets. On the top and right, the density plot of $\log_2$(cyto/nuc) expression changes is depicted. (**B**) Quantification of $\log_2$(cyto/nuc) expression changes in iKO vs. Ctrl. (**C**) Overlapping between YTHDC1 mRNA export targets and splicing targets in ASCs. (**D**) Genome tracks of two selected export targets. (**E–I**) The

*Figure 6 continued on next page*

*Figure 6 continued*

above assay/analysis was performed in DMSO or IAA-treated mAID-YTHDC1 C2C12 myoblasts to identify YTHDC1 regulated export targets in C2C12. (**J–N**) The above analyses were conducted using m6A-YTHDC1 targets to identify m6A-YTHDC1 mRNA export targets in C2C12.

The online version of this article includes the following figure supplement(s) for figure 6:

**Figure supplement 1.** Analyses of nuclear export of all mRNAs and GO enrichment of mRNAs with decreased export.

partners of YTHDC1, SRSF3 (*Xiao et al., 2016*) on the list (*Figure 7E*), strengthening its function in regulating splicing in myoblasts.

Examining the list of mRNA export factors (*Figure 7F*), several components of the THO subcomplex of the TREX (transcription export) complex were identified. The bulk of the mRNA nuclear export is a complex yet well-characterized process mediated by the TREX complex and the heterodimeric nuclear export receptor NXF1:NXT1 (*Lesbirel et al., 2018*; *Lesbirel and Wilson, 2019*). As the key initiating complex, TREX has 14 known subunits with the multimeric THO complex as the core. Strikingly, all the known THOC members, including THOC7, 1, 6, 3, 5, and 2, were retrieved by YTHDC1 (*Figure 7F*). We validated the interaction of YTHDC1 with THOC7 in C2C12 by co-IP followed by WB (*Figure 7L*), reinforcing YTHDC1 function in controlling mRNA export. Furthermore, we also identified some known transcriptional regulators, including RTRAF, MED29, and MED11 (*Figure 7G*), pointing to previously unknown means via which YTHDC1 may regulate transcription. Altogether, the results from the co-IP/MS are in line with the findings from *Figures 5 and 6* to reinforce the notion that YTHDC1 plays pleiotropic regulatory functions in myoblasts.

## Discussion

In this study, we investigate the functional role of m6A reader YTHDC1 protein in skeletal muscle stem cells and muscle regeneration. Our findings demonstrate the expression dynamics of several m6A regulators including writers, readers, and erasers during the course of SC lineage progression, implicating their possible involvement in governing SC activities. Among these m6A regulators, we characterize YTHDC1 function in depth and uncover it as an essential factor controlling SC activation and proliferation. Inducible depletion of YTHDC1 in SCs drastically impairs SC activation and proliferation; thus, it almost abolishes acute injury-induced muscle regeneration. Mechanistically, LACE-seq identifies YTHDC1 binding mRNA targets among which a portion are m6A dependent. Further splicing analyses provide evidence for m6A-YTHDC1 participation in modulating splicing events in myoblast cells. Additionally, subcellular fractionation/RNA-seq also defines potential mRNA export targets of YTHDC1 in myoblasts. Lastly, co-IP/MS defines a wide array of interacting protein partners of YTHDC1, including mRNA splicing and export factors, as well as transcriptional regulators, and they may function in synergism to mediate the pleiotropic functions of YTHDC1 in myoblasts. Among these factors, hnRNPG appears to be a bona fide functional partner of YTHDC1 to synergistically regulate myoblast proliferation. Altogether, our findings demonstrate that YTHDC1 is an indispensable intrinsic regulator of SC activities and muscle regeneration through multifaceted controlling of RNA metabolism in myoblast cells (*Figure 8*).

Until now the study of m6A regulation and function in SCs is scarce (*Diao et al., 2021*; *Kudou et al., 2017*; *Liang et al., 2021*). This is the first study to identify potential functional m6A regulators and provides a holistic investigation of a writer protein, YTHDC1 function in SCs and muscle regeneration. The transcriptomic profiling uncovered that an array of m6A regulators were dynamically expressed in the course of SC lineage progression; notably, YTH-domain containing reader proteins, including YTHDC1, YTHDF1, YTHDF2, and YTHDF3, were all highly expressed in SCs. The mRNA level of YTHDC1 was high across the entire course albeit showing an induction in FISC; its protein level, nevertheless, was induced in ASC-24 hr and continued to increase until ASC-48 hr, suggesting its dominant role in governing activation/proliferation. The discrepancy between mRNA and protein expression also suggests that there may be post-transcriptional regulation to control YTHDC1 protein production. It is possible that the induction of *Ythdc1* mRNA occurs early upon activation, but the protein translation occurs later in myoblasts, where it executes the mRNA splicing and export functions. Indeed, the subsequent dissection using the YTHDC1-iKO mouse provided solid evidence supporting its positive role in promoting SC activation and proliferation; inducible depletion of YTHDC1 largely inhibited SC activation and proliferation (*Figure 3*). We believe the incompetence

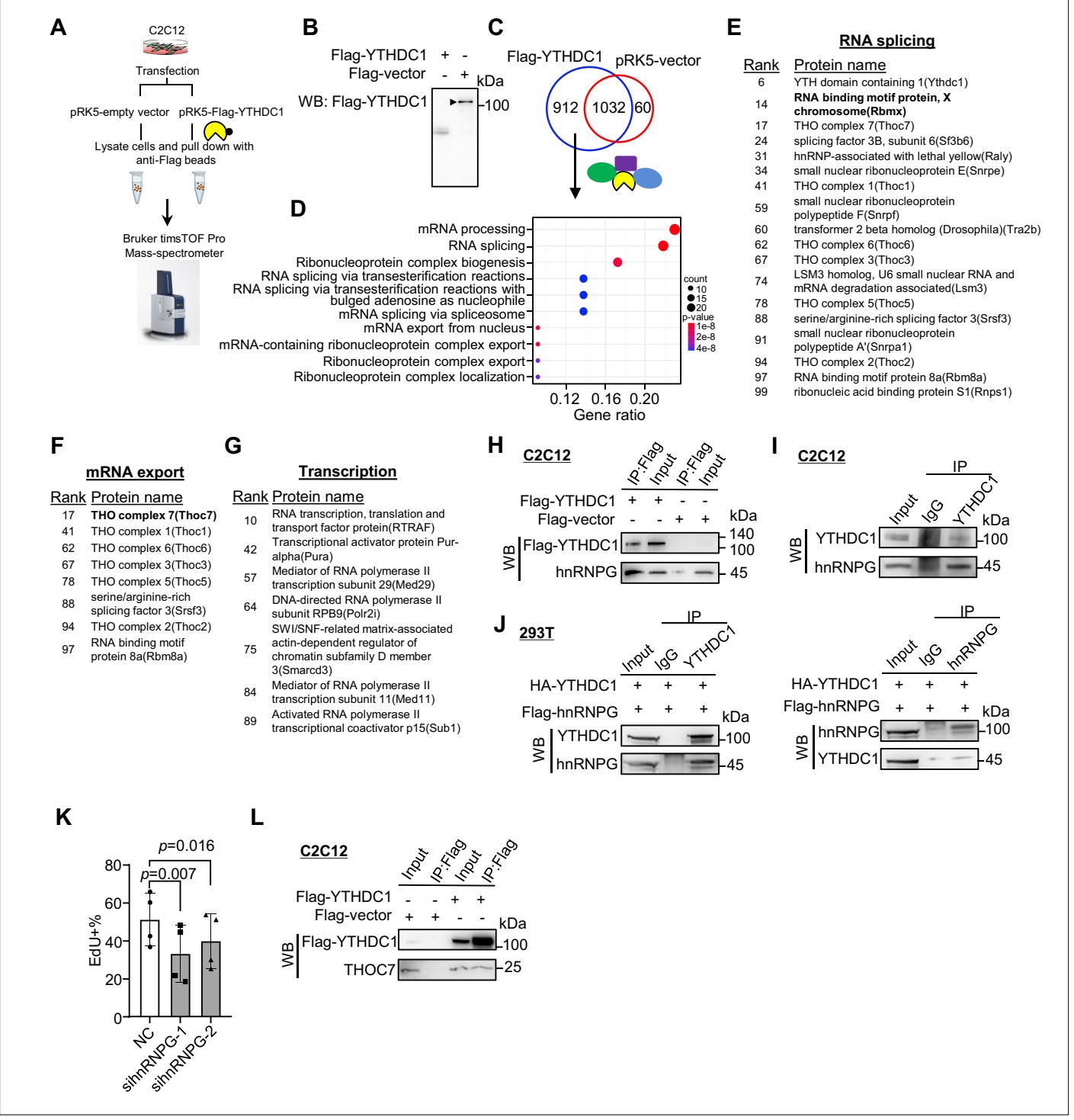

**Figure 7.** Co-immunoprecipitation/mass spectrometry (co-IP/MS) leads to the identification of YTHDC1 interacting partners. (**A, B**) Schematic illustration of the co-IP/MS procedure. An empty vector or Flag-tagged YTHDC1 plasmid was expressed in C2C12 myoblasts followed by pull-down with Flag beads; the retrieved proteins were subject to MS with Bruker timsTOF Pro. (**B**) Overexpression of the Flag-tagged YTHDC1 was confirmed by western blotting (WB) using anti-Flag antibody. (**C**) 912 proteins were uniquely retrieved in YTHDC1 but not Vector-expressing cells. (**D**) GO functions of the above proteins are shown. (**E–G**) The above-identified interacting proteins with RNA splicing, mRNA export, or potential transcriptional regulatory functions are shown in the lists. (**H**) Flag-tagged YTHDC1 was overexpressed in C2C12 and Flag-beads-based IP was performed followed by WB to verify the retrieved hnRNPG protein. (**I**) IP of endogenous YTHDC1 protein in C2C12 myoblasts followed by WB to examine retrieved hnRNPG protein. (**J**) Flag-hnRNPG and HA-YTHDC were overexpressed in 293T cells; IP with YTHDC1 or hnRNPG protein flowed by WB to confirm the interaction

*Figure 7 continued on next page*

*Figure 7 continued*

between the two proteins. (**K**) Freshly isolated satellite cells (FISCs) were transfected with scramble or hnRNPG siRNA-1 or -2, and EdU assays were performed in ASC-48 hr. EdU+ cells were quantified. n = 4 replicates. Bars represent mean ± SD for all graphs. Statistical significance was determined using a two-tailed paired Student's *t*-test. (**L**) Flag-tagged YTHDC1 was overexpressed in C2C12 and Flag beads-based IP was performed followed by WB to verify the interaction between YTHDC1 and THOC7.

The online version of this article includes the following source data for figure 7:

**Source data 1.** Uncropped blot images of *Figure 7B and H*.

**Source data 2.** Uncropped blot images of *Figure 7I and J*.

**Source data 3.** Uncropped blot images of *Figure 7L*.

of SC activation/proliferation is the major cause of nearly blocked regeneration after acute injury-induced muscle damage. Nevertheless, the potential defects in other aspects such as differentiation, self-renewal, and survival need to be investigated in the future. Of note, YTHDC1 expression can also be detected in QSCs albeit to a very low level. We also noticed decreased SC numbers after long-term loss of YTHDC1 (data not shown), suggesting that it may play a critical role in QSC maintenance. Our findings thus suggest that YTHDC1 plays an indispensable role in SCs and muscle regeneration, which is not replaceable by other m6A readers. This study also adds genetic evidence for the importance of YTHDC1 in cellular processes, which is largely lacking in the field. In the future, it will also be interesting to test the functionality of other readers such as YTHDF1, 2, and 3 in SCs.

To investigate the functional mechanism of YTHDC1 in promoting SC activation/proliferation, recently developed LACE-seq was harnessed using both ASCs and C2C12 myoblasts. Even though the number of peaks identified in ASCs was not comparable with the number in C2C12, LACE-seq

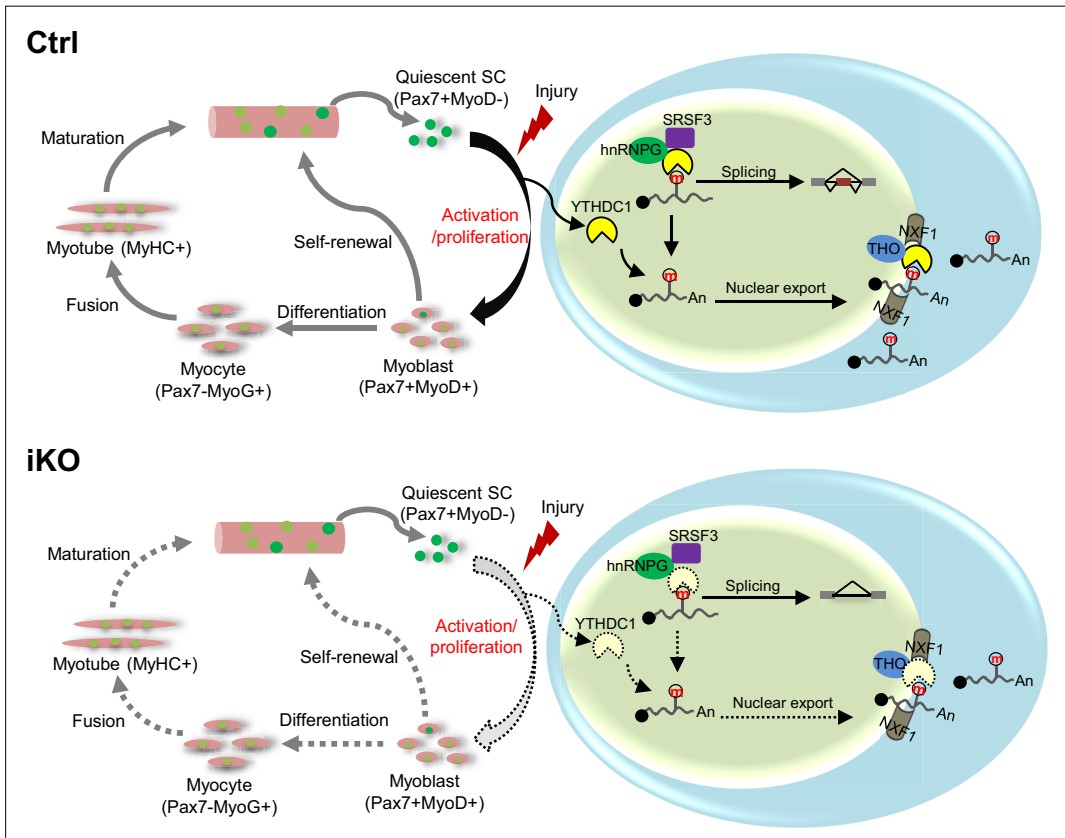

**Figure 8.** A working model of YTHDC1 function in satellite cells (SCs). YTHDC1 is induced upon SC activation and proliferation to promote SC activation and proliferation upon acute injury-induced muscle regeneration. Mechanistically, it functions through both facilitating mRNA splicing synergistically with hnRNPG and promoting mRNA export possibly by binding with the THO nuclear export complex.

has enabled the profiling of transcriptome-wide YTHDC1 binding in ASCs, which was previously impossible using traditional mRNA binding probing methods. By combining with the m6A MeRIP-seq dataset available in C2C12 (*Gheller et al., 2020*), we were able to define m6A-modified binding targets of YTHDC1. From out data, YTHDC1 selectively bound a subset but not all m6A-modified RNAs (*Figure 4H and J*). This selectivity could be achieved by dynamic multivalent interactions of YTHDC1, RNAs, and regulatory proteins that self-organize into distinct condensates (nuclear subcompartments) within the nucleus (*He and He, 2021*; *Widagdo et al., 2022*), It is also worth pointing out that a large portion of the YTHDC1 binding targets do not seem to possess m6A modification (*Figure 4H*), suggesting that YTHDC1 may have m6A-independent functions. However, it is commonly believed that YTHDC1 regulatory functions are largely dependent on its m6A binding capacity (*Chen et al., 2021a*; *Cheng et al., 2021*; *Sheng et al., 2021*; *Xiao et al., 2016*). In *Figure 3J*, the mutants also failed to rescue the myoblast proliferative defect in YTHDC1-depleted C2C12 myoblasts, strongly suggesting its pro-proliferative function in myoblasts relies on its m6A binding ability. Nevertheless, since YTHDC1 is now known to participate in liquid phase separation (*Widagdo et al., 2022*), it is likely that m6A-modified RNAs may initiate the recruitment of YTHDC1 through the YTH m6A binding domain, YTHDC1 then promotes condensate formation through the two intrinsically disordered regions (IDRs) independent of its m6A reading function. It is also interesting to notice that about 12% of YTHDC1 binding peaks were mapped to retrotransposons in ASCs and 10% in C2C12 cells (*Figure 4F and G*). In mESCs, YTHDC1 silences retrotransposons to guard ES cell identity (Chen et al., 2021a; *Liu et al., 2021*; *Xu et al., 2021*), it would thus be interesting to explore the possible functional relevancy of YTHDC1 regulation of retrotransposons in SCs. In the future, it will also be necessary to generate transcriptome-wide m6A mapping in SCs, which has become possible by the recently developed methods enabling the use of low number of cells.

Next, we examined YTHDC1 regulation of mRNA splicing in depth by examining altered splicing events upon YTHDC1 loss, which unveiled the splicing targets of m6A-YTHDC1. The number was relatively small in ASCs due to the relatively small number of YTHDC1 binding targets from LACE-seq; a much higher number was found in C2C12 cells but constituted only a small number (20%) of m6A-YTHDC1 targets. Additionally, we examined the potential effects of YTHDC1 loss in mRNA nuclear export and indeed identified a set of mRNAs possibly regulated by m6A-YTHDC1; upon YTHDC1 loss, the nuclear enrichment level was significantly increased. Interestingly, 20% of these mRNAs are also splicing targets of m6A-YTHDC1, suggesting that YTHDC1 binding to an m6A-modified mRNA can control mRNA metabolism at multiple levels. Of note, our study identifies a large number of splicing or/and export targets of m6A-YTHDC1, and they play a wide range of cellular functions directly or indirectly related to cellular proliferation. The observed proliferative defects upon YTHDC1 loss thus could be a combinatorial effect of many downstream targets that were enriched for a wide range of GO functions important for SC proliferation (*Figure 5—figure supplement 1A–D* and *Figure 6G and L*). For example, canonical Wnt signaling is known to induce SC proliferation during adult skeletal muscle regeneration (*Otto et al., 2008*).

Lastly, co-IP/MS was performed in myoblasts and led to the identification of an array of RNA processing regulators mainly including splicing and nuclear export factors. This finding is in concordance with the above demonstrated YTHDC1 functions in splicing and export regulations. In addition, transcriptional regulators were also identified. The m6A-dependent functions of YTHDC1 in regulating splicing, export, or transcription have all been individually reported in various cell types (*Kasowitz et al., 2018*; *Lee et al., 2021*; *Roundtree et al., 2017*); our finding thus hints pleiotropic roles of YTHDC1 in myoblasts. In terms of splicing regulation, Xiao et al. showed that YTHDC1 binds competitively to the splicing factor SRSF3, while antagonizing the binding of SRSF10 to promote exon inclusion (*Xiao et al., 2016*). The top-ranked binding partner on our list, however, was not SR splicing factors (*Figure 7E*); instead, hnRNPG appeared as the most likely interacting partner. Its specific interaction with YTHDC1 was validated by co-IP/WB using both exogenously and endogenously expressed proteins in C2C12 and also in 293T cells (*Figure 7*), thus pointing to hnRNPG as a bona fide interacting partner of YTHDC1. Interestingly, hnRNPG also represents a multifaceted gene regulator. Originally known as a hnRNP family splicing factor via interactions with SRSF3, SRSF7, SLM, SAFB1 (*Elliott et al., 2019*; *Heinrich et al., 2009*), it was later shown to bind with m6A-modified mRNAs, thus possessing m6A reader function. It can co-transcriptionally interact with both RNAPII and m6A-modified nascent pre-mRNA to modulate RNAPII occupancy and alterative splicing (Zhou

et al., 2019). Its implicated role in transcription repression was also reported (*Elliott et al., 2019*). Therefore, both hnRNPG and YTHDC1 are versatile m6A-dependent gene regulators yet no study so far has made functional connection between these two key m6A readers despite their physical interaction being mentioned (*Heinrich et al., 2009*; *Xu et al., 2022*). Our result in *Figure 7K* demonstrates their functional synergism in promoting C2C12 myoblast proliferation. Future efforts will be focused on further delineating the regulatory and functional synergism between YTHDC1 and hnRNPG in SCs. In addition, the co-IP/MS also identifies a number of TREX components as potential interacting partners of YTHDC1, and a prior report also demonstrated the physical interaction between TREX complex with YTHDC1 in HEK293T cells (*Lesbirel et al., 2018*). The modulation of nuclear export by YTHDC1 is thus very likely exerted together with TREX proteins, which will need to be further investigated in the future.

Altogether, biologically, our findings uncover YTHDC1 as an essential post-transcriptional regulator of SC activities and muscle regeneration. In the coming years, more solid genetic evidence will be needed to demonstrate the key roles of m6A regulators in various biological systems. It will also be interesting to determine whether YTHDC1 deregulation is implicated in muscle-related diseases such as aging-associated sarcopenia. Mechanistically, we demonstrate multifaceted roles of YTHDC1 in decoding m6A-marked transcripts in myoblasts; a wide range of mRNA targets are controlled by YTHDC1, thus mediating its effect in promoting myoblast proliferation. Although this study mainly focuses on delineating its post-transcriptional actions, we believe that YTHDC1 also modulates transcriptional events in myoblast nucleus and hnRNPG seems to constitute an important co-factor that may synergistically regulate both splicing and transcription functions of YTHDC1. In addition, it will be interesting to explore whether YTHDC1, hnRNPG, and the target mRNAs self-organize into distinct condensates within the myoblast nucleus as emerging evidence suggests RNA, m6A readers, and co-factors can cooperatively form distinct nuclear condensates at specific genomic loci (*Widagdo et al., 2022*). Furthermore, although YTHDC1 is predominantly found in the nucleus of myoblast cells, it may transiently shuttle into the cytoplasm as described in *Rafalska et al., 2004*; *Shima et al., 2017* and execute unexpected post-transcriptional functions such as in mRNA destabilization (*Shima et al., 2017*; *Zhang et al., 2020*).

# Materials and methods

## Mice

*Pax7^{CreER}* (*Pax7^{tm1(cre/ERT2)Gaka}*) and Tg: Pax7-nGFP mouse strains were kindly provided by Dr. Shahragim Tajbakhsh. *Ythdc1^{fl}*(*B6;129S4-Ythdc1tm1.1Jw/Mmjax*) strain (050693-JAX) and *Rosa26^{EYFP}* mouse (006148-JAX) were provided by Jackson Laboratory. *Pax7^{CreER}* mice were crossed with *Rosa26^{EYFP}* mice to generate the *Pax7^{CreER}*; *Rosa26^{EYFP}* reporter mice. The *Ythdc1* inducible knock out (*Ythdc1* iKO) mice with EYFP reporter (Ctrl: *Pax7^{CreER/+}*; *Rosa26^{EYFP/+}*; *Ythdc1^{+/+}*, iKO: *Pax7^{CreER/+}*; *Rosa26^{EYFP/+}*; *Ythdc1^{fl/fl}*) were generated by crossing *Ythdc1^{fl}* mice with *Pax7^{CreER}*; *Rosa26^{EYFP}* reporter mice. The mice were maintained in animal room with 12 hr light/12 hr dark cycles, 22–24°C room temperature (RT) and 40–60% humidity at the animal facility in the Chinese University of Hong Kong (CUHK). All animal handling procedures and protocols were approved by the Animal Experimentation Ethics Committee (AEEC) of CUHK (ref. no. 19-251-MIS). All animal experiments with iKO mice followed the regulations and guidance of laboratory animals set in CUHK.

## Animal procedures

Inducible conditional deletion of *Ythdc1* was administered by injecting tamoxifen (Tmx) (T5648, Sigma) intraperitoneally (IP) at 2 mg per 20 g body weight. For $BaCl_2$-induced muscle injury, 2–3-month-old mice were intramuscularly injected with 50 µl of 1.2% $BaCl_2$ solution into TA muscles, and the muscles were harvested at designated time points for further analysis. For EdU incorporation assay in vivo, 2 days after $BaCl_2$ injection, EdU was injected via IP at 0.25 mg per 20 g body weight, followed by FACS isolation of SCs 12 hr later. SCs were then seeded and fixed with 4% PFA for further stain and analysis.

## Satellite cell isolation by FACS

Briefly, hindlimb muscles from *Ythdc1* Ctrl/iKO and Pax7n-GFP mice were dissected and minced with blades, then digested with collagenase II (1100 U/ml, Worthington in Hams F-10 media [Sigma]) for

90 min at 37°C with gentle rotation at 70 rpm. The digested muscles were washed in washing medium (Hams F-10 media, 10% HIHS [Gibco], penicillin/streptomycin [1×, Gibco]) once and SCs were further released by treating muscles with collagenase II (1100 U/ml) and dispase (11 U/ml) for 40 min at 37°C. Digested tissue was passed through a 21-gauge needle 12 times and filtered through a 40 µm filter followed by spinning at 700 × *g* for 5 min at 4°C. Mononuclear cells were resuspended and filtered with a 40 µm cell strainer and GFP+/EYFP+ SCs were sorted out by BD FACS Aria Fusion cell sorter (BD Biosciences).

### Single myofiber isolation

Single myofibers were isolated as previous described (*Chen et al., 2021b*). 2–3-month-old mice were used for extensor digitorum longus muscles (EDL) isolation. After incubation in collagenase II solution (800 U/ml) at 37°C in a water bath for 70 min, muscles were transferred to a dish containing prewarmed washing medium (F10 + 10% horse serum + penicillin-streptomycin). Single myofibers were released by pipetting with a large hole bore glass pipette gently and transferred to a new dish with culture medium (F10 + 10% horse serum + penicillin-streptomycin + b-FGF [0.025 µg/ml]) for ex vivo culture. Fibers were fixed with 4% PFA at designed time point. For EdU assay, 10 µM EdU was added to culture medium for 4 hr before fixation.

### Cell lines and cell culture

Mouse C2C12 myoblast cells (CRL-1772) and 293T cells (CRL-3216) were obtained from American Type Culture Collection (ATCC) and cultured in DMEM medium with 10% fetal bovine serum, 100 units/ml of penicillin, and 100 µg of streptomycin (growth medium [GM]) at 37°C in 5% $CO_2$. All cell lines were tested as negative for mycoplasma contamination. Freshly isolated SCs were cultured in Ham's F10 medium supplemented with 20% FBS and bFGF (0.025 µg/ml) (GM) on dish/slides precoated with PDL and ECM, ASC-24 hr and ASC-48 hr were harvested at designed time point for RNA/protein/immunofluorescence analysis.

### Generation of mAID-YTHDC1-inducible degradation C2C12 cells

Briefly, (*Yesbolatova et al., 2019*) OsTIR1 (F74G) mutant was first introduced into PB-Ostir-neo (Addgene plasmid #161973) plasmid by overlap PCR to generate AID2 system in C2C12 cells with improved degradation efficiency and reduced background degradation. CRISPR–Cas9 plasmid was generated using PX458 (Addgene plasmid #48138) and sgRNA targeting N-terminal of YTHDC1 near ATG start codon. Two donor plasmids were generated by overlap PCR ~ 500 bp *Ythdc1* genomic sequence flanking BSD/HygR-P2A-mAID-mCherry2 sequence (Addgene plasmid #121180, #121183) and cloning into pMD20-T vectors by T-A ligation. Cas9 and donor plasmids were transfected to C2C12 (Ostir2) cells and antibiotic selection (BSD 10 ug/ml, Hygro 100 ug/ml) was performed for 1 week before single clone selection. Successful knocked-in clones were validated by genotyping PCR and western blot. 1 µM 5-Ph-IAA was used to induce mAID-YTHDC1 degradation with DMSO treatment used as a control.

### EdU assay

EdU assay was performed following the manufacturer's protocol (Invitrogen, Click-iT EdU Cell Proliferation Kit for Imaging, Alexa Fluor 594 dye, C10339). Cells/myofibers were incubated with 10 µM EdU for 4 hr before fixation with 4% PFA.

### Plasmids

pRK5-flag-YTHDC1 and pRK5-flag hnRNPG plasmids were generated by amplifying ORF of YTHDC1 and hnRNPG from SCs cDNA and cloned into pRK5-flag empty vector through NotI and HindIII restriction sites. pRlenti-HA-YTHDC1, pRlenti-W429A-YTHDC1, and the pRlenti-triple-mutant-YTHDC1 plasmids were gifts from Prof. Jiekai Chen. pRK5-W429A-YTHDC1 and pRK5-triple-mutant-YTHDC1 were generated by amplifying ORF of YTHDC1 from the related pRlenti plasmids and cloned into pRK5-flag empty vector.

### Cell fractionation

Cell fractionation protocol was modified based on previous protocol (*Huang et al., 2021*). C2C12 or SCs were collected in cold PBS, washed once, and then incubated in buffer A (HEPES-KOH 50 mM pH

7.5, 10 mM KCl, 350 mM sucrose, 1 mM EDTA, 1 mM DTT, 0.1% Triton X-100) for 5 min on ice and then homogenized with a T 10 basic ULTRA-TURRAX homogenizer at fourth gear for 1 min. The nuclei were harvested by brief centrifugation (2000 × $g$, 5 min), while the supernatant was collected as the cytoplasmic fraction. Nuclei were resuspended with same volume buffer A as supernatant. RNAs were extracted using TRIzol reagent and further analyzed by RT-qPCR and RNA-seq. For proteins, 6× SDS loading buffer was added accordingly for western blot analysis.

## RNA isolation and quantitative RT-PCR

Total RNAs were extracted using TRIzol reagent (Invitrogen) following the manufacturer's protocol. For quantitative RT-PCR, cDNAs were reverse transcribed using HiScript III First-Strand cDNA Synthesis Kit (Vazyme, R312-01). Real-time PCR reactions were performed on a LightCycler 480 Instrument II (Roche Life Science) using Luna Universal qPCR Master Mix (NEB, M3003L). For splicing verification, GAPDH was used as a control. cDNA of Ctrl/iKO were amplified by 30 cycles of PCR and run on 2% agarose gel. Sequences of all primers used can be found in *Supplementary file 6*.

## MeRIP-qPCR

C2C12 cells were treated with DMSO or 40 µM STM2457 (MCE, Cat# HY-134836) for 24 hr. Cells were washed with PBS once and RNAs were harvested and extracted using TRIzol. 2.5 µg RNA was diluted in 250 µl IP buffer (10 mM Tris-HCl, pH 7.4, 150 mM NaCl, 0.1% NP-40, and Protease Inhibitor Cocktail [Sigma-Aldrich]). Then, 50 µl of diluted RNA was saved as 25% input. And 20 µl Dynabeads protein G (Thermo Fisher/Life Technologies) was washed three times with 1 ml IP buffer and incubated with 2 µg m6A antibody (Abcam, ab151230) at RT for 30 min. The conjugated Dynabeads-m6A-antibody was washed in 1 ml IP buffer three times and resuspended in 800 µl IP buffer. The 200 µl RNA sample was mixed with antibody-protein G beads and rotated at RT for 3 hr. After 3 hr rotation, the beads were washed with IP buffer five times and resuspended in 50 µl IP buffer. Both immunoprecipitated and input RNAs were extracted from the beads by TRIzol extraction. The extracted RNA was subjected to cDNA synthesis and qPCR to quantify the m6A levels. Sequences of all primers used can be found in *Supplementary file 6*.

## RIP-qPCR

C2C12 cells cultured in 10 cm dish were transfected with 10 µg pRK5-flag-YTHDC1 using Lipofect-amine 3000 reagent (Invitrogen). Then, 24 hr after transfection, the medium was changed and DMSO or 40 µM STM2457 were added for additional 24 hr. Cells were washed twice with ice-cold PBS and 3 ml cold PBS were added to the dish. Also, 400 mJ/cm² UV was used to crosslink RNAs with interacting proteins twice and cells were collected by scraping and centrifugation. Cells were lysed in 1 ml RIPA buffer (50 mM Tris, pH 7.4, 150 mM NaCl, 1 mM EDTA, 0.1% SDS, 1% NP-40, 0.5% sodium deoxycholate, 1 mM DTT, Protease Inhibitor Cocktail, 0.4 U/µl RNaseOUT Recombinant Ribo-nuclease Inhibitor, Invitrogen) at 4°C for 25 min with rotation. Lysate was sonicated (SONICS, VCX130, 1 min on/1 min off for five cycles) and centrifuged at 16,000 × $g$ for 15 min. Then, 10 µl flag beads (MCE, Cat# HY-K0207) were added to the supernatant. Also, 800 µl was used for IP, 100 µl was saved for RNA input, and 50 µl was saved for western blot input. After incubation at 4°C overnight with rotation, IP beads were washed five times with RIPA buffer and resuspended in 100 µl RIPA buffer (without Protease Inhibitor Cocktail). Both input and IP samples were digested with 2.5 µl proteinase K (20 mg/ml, Ambion, AM2456) at 55°C for 30 min. RNA was recovered by TRIzol reagent and reverse transcribed using HiScript III First-Strand cDNA Synthesis Kit (Vazyme, R312-01) with an additional genomic DNA remove step. Then the IP enrichment of YTHDC1-m6A targets was examined by qPCR.

## RNA-seq and data analysis

For RNA-seq (polyA+mRNA), total RNAs were subject to polyA selection (Ambion, 61006) followed by library preparation using NEBNext Ultra II RNA Library Preparation Kit (NEB, E7770S). Libraries were paired-end sequenced with read lengths of 150 bp on Illumina HiSeq X Ten or Nova-seq instru-ments. The raw reads of RNA-seq were processed following the procedures described in our previous publication. Briefly, the adapter and low-quality sequences were trimmed from 3′ to 5′ ends for each read, and the reads shorter than 50 bp were discarded. The clean reads were aligned to mouse (mm9) reference genome with STAR. Next, we used Cufflinks to quantify the gene expression. Genes

were identified as DEGs if the change of expression level is greater than twofold and the p-value is <0.01 between two stages/conditions. GO enrichment analysis were performed using R package clusterProfiler.

## Immunoblotting and immunofluorescence

Immunoblotting and immunofluorescence were performed according to our standard protocols (*Chen et al., 2019*; *Chen et al., 2021b*; *Li et al., 2020*; *Zhao et al., 2019*). Proteins were extracted using RIPA lysis buffer. The following dilutions of antibodies were used for western blot staining: PAX7 (Developmental Studies Hybridoma Bank; 1:200), GAPDH (Sigma, G9545, 1:3000), YTHDC1 (Cell Signaling Technology, #77422, 1:1000), MyoD (Abclonal, 1:1000), hnRNPG (Cell Signaling Technology, #14794, 1:1000), histone H3 (Santa Cruz, sc-517576, 1:5000), Flag-tag (Sigma, F3165, 1:3000), and THOC7 (Abclonal, A13700, 1:1000). For immunofluorescence staining, cultured cells and myofibers were fixed with 4% PFA at RT for 15 min, after washing with PBS, cells were permeabilized with 0.2% TritonX-100 and followed by 30 min blocking with 3% BSA. Primary antibodies were incubated at 4°C overnight. Following antibodies and related dilutions were used: PAX7 (Developmental Studies Hybridoma Bank; 1:50), eMyHC (Leica NCL-MHC-d;1:200), laminin (Sigma-Aldrich L9393; 1:800), MyoD (Abclonal, 1:1000), and YTHDC1 (Abcam, ab122340, 1:200). Secondary antibodies were incubated at RT for 1 hr, and nuclei were stained with ProLong Gold Antifade Mountant with DAPI (Thermo). Immunofluorescence staining of frozen muscle sections was performed as previously described (*Chen et al., 2019*). Briefly, samples were boiled in 0.01 M citric acid (pH 6.0) for 10 min in a microwave before blocking with 4% BBBSA (4% IgG-free BSA in PBS; Jackson, ref: 001-000-162). Then the endogenous mouse IgG were blocked by treatment with the donkey anti-mouse IgG (H+L) (1/100 in PBS; Jackson, ref: 115-007-003) for 30 min. After primary antibody incubation overnight, the biotin-conjugated anti-mouse IgG (1:500 in 4% BBBSA, Jackson, ref: 205) and Cy3-Streptavidin (1:1250 in 4%BBBSA, Jackson, ref: 016-160) were used as secondary antibodies for Pax7 staining. All fluorescent images were captured with a fluorescence microscope (Leica DM6000 B). H&E staining on frozen muscle sections was performed as previously described (*Li et al., 2020*).

## Co-IP/mass spectrometry

Immunoprecipitation of C2C12 cell overexpressing flag-YTHDC1 was performed based on previous publication (*Zhao et al., 2019*). Briefly, C2C12 cells were transfected with pRK5-flag-YTHDC1 or pRK5-falg empty vector for 48 hr. Cells were digested with trypsin and washed with PBS once, then lysed in hypotonic lysis buffer (10 mM HEPES, pH 7.9, 10 mM KCl, 0.1 mM EDTA, 0.1 mM EGTA, and cOmplete Protease Inhibitors [Roche]) and incubated on ice for 15 min. The lysates were centrifuged for 10 min at 2000 × g and supernatant was discarded. The pelleted nuclei were washed once with hypotonic lysis buffer and then resuspended in hypertonic buffer (20 mM HEPES, pH 7.9, 0.4 M NaCl, 1 mM EDTA, 1 mM EGTA, 0.6% NP-40, and cOmplete Protease Inhibitors [Roche]), digested with the DNase I (AM2238, Thermo Fisher Scientific) for 45 min at 4°C, and spun down at 12,000 × g for 10 min at 4°C. The nuclear lysates were diluted twofold with IP buffer (20 mM HEPES, pH 7.9, 0.2 M NaCl, cOmplete Protease Inhibitors [Roche]) and then flag-beads (Anti-Flag Magnetic Beads, MCE, HY-K0207) were added and incubated overnight at 4°C. After washing five times, proteins were eluted with elution buffer (62.5 mM Tris-HCl 7.5, 0.2% SDS) at 99°C for 10 min, and 20% elution was subjected to western blot verification. Mass spectrometry experiment was performed using the Bruker timsTOF Pro Mass-spectrometer with the help of Biosciences Central Research Facility, HKUST. Mass spectrometry raw data was processed by PEAKS software (version X+). Protein abundance was obtained by normalizing the spectral number of proteins with the length of the protein. Unique YTHDC1 interacting proteins were selected comparing flag-YTHDC1 sample and flag-vector sample (excluding nonspecific binding targets).

## Co-immunoprecipitation

Co-IP of C2C12 cells and 293T cells was performed based on published protocol (*Xu et al., 2021*). Briefly, cells were incubated in buffer A for 5 min on ice and then homogenized as described for cell fractionation. The nuclei were pelleted by 2000 × g, 5 min centrifugation and resuspended in hypertonic buffer (20 mM HEPES, pH 7.5, 10% glycerol, 0.42 M KCl, 4 mM MgCl$_2$, 0.2 mM EDTA, 0.5 mM DTT, Protease Inhibitor Cocktail). After 30 min incubation on ice, nuclear extract was collected

by high-speed centrifugation (12,000 rpm, 15 min, 4°C). Same volume of hypotonic buffer (10 mM HEPES pH 7.5, 1.5 mM MgCl$_2$, 10 mM KCl, 0.5 mM DTT, 1× Protease Inhibitor Cocktail) was added to nuclear extract. Lysate was precleared with protein G beads (Dynabeads Protein G for Immunoprecip-itation, Invitrogen, 10003D) at 4°C for 1 hr with rotation and 10% was saved as input. After incubation with indicated antibodies overnight, protein G beads were added and incubated for another 2 hr at 4°C. Beads were washed five times with IP buffer and proteins were eluted with 1× SDS loading buffer at 99°C for 5 min.

## LACE-seq and data analysis

LACE-seq was performed as described. (*Su et al., 2021*) Cells were washed with ice-cold PBS and subjected to 400 mJ UV treatment to crosslink proteins with their interacting RNAs. After cross-link, cells were pelleted and kept at –80°C before library preparation. YTHDC1 antibody (Abcam, ab122340) was used to pull down specific protein-RNA complex from lysate. In brief, MNase was used to cut RNAs into YTHDC1-associated short fragments on beads. The 3′ ends of fragmented RNAs were then dephosphorylated and ligated with a 5′ pre-adenylated linker containing four randomized nucleotides. Biotinylated primer containing the T7 promoter was used for reverse transcription, which stopped at crosslinking site (YTHDC1 binding site). cDNAs were poly(A) tailed and purified with streptavidin beads. Second-strand cDNAs were synthesized on beads with an adaptor containing oligo-(dT). After a pre-amplifying step by PCR, T7 RNA polymerase was used to amplify trace amounts of truncated cDNAs linearly. Then the products were PCR converted into libraries for single-end sequencing on Illumina platform (NextSeq 550). For the LACE-seq data analysis, first, the adapter sequences and poly(A) tails at the 3′ end of raw reads were removed using Cutadapt (v.1.15) with the following parameters: -f fastq -m 18 -n 2 -a A{15} `--quality-base=33`. Clean reads were first aligned to mouse pre-rRNA using Bowtie software (v.1.2.3) with default parameters, and the remaining unmapped reads were then aligned to the mouse (mm9) reference genome with Bowtie parameters: `--best --strata` -v 2 -k 10. Pearson's correlation coefficient between LACE-seq replicates was performed using multiBamSummary module of deeptools. LACE-seq peaks are called by Piranha software (http://smithlabresearch.org/software/piranha/, v.1.2.1) in ASC cells. The parameters were as follows: -sort -p_threshold 0.001 -b 20 -d ZeroTruncatedNegativeBinomial. For motif analysis, LACE-seq peaks were first extended 20 nt to upstream and downstream, respectively. Enriched motifs are scanned by findMotifsGenome.pl from Homer.

## Whole-cell RNA-seq and RNA splicing analysis

For the splicing analysis, RNA-seq reads were first aligned to the mm9 reference genome using STAR, and splicing events were then detected by rMATS with default parameters (*Shen et al., 2014*). A cutoff of absolute value of 'IncLevelDifference' < 0.1 was used to define DSEs in YTHDC1 iKO vs. Ctrl. DSE targets are the genes that having DSE in its transcripts.

## Subcellular RNA-seq and mRNA exporting analysis

For the subcellular RNA-seq analysis, reads were aligned to reference genome and the quantification of gene expression was performed by Cufflink. A cutoff of FPKM > 1 either in cytoplasmic (cyto) or nuclear (nuc) was used to determine the expression of genes. To identify the nuclear-enriched mRNAs upon YTHDC1 knockout or degradation, the $\log_2$(cyto/nuc) of knockout or degradation should be less than the Ctrl sample and the $\log_2$(cyto/nuc) of knockout or degradation should be less than –1.

## Acknowledgements

We thank Dr Lifang Han for assisting us in performing the Mass Spectrometry at Biosciences Central Research Facility, HKUST; Professor Jun Yu and Professor Huarong Chen for the help with the METTL3 inhibitor STM2457.

## Additional information

### Funding

| Funder | Grant reference number | Author |
|---|---|---|
| National Key R&D Program of China | 2022YFA0806003 | Huating Wang |
| National Natural Science Foundation of China | 82172436 | Huating Wang |
| Health@InnoHK program launched by Innovation Technology Commission, the Government of the Hong Kong SAR, China | Center for Neuromusculoskeletal Restorative Medicine (CNRM) | Huating Wang |
| Research Grants Council of Hong Kong | Theme-based Research Scheme (TRS) T13-602/21-N | Huating Wang |
| Research Grants Council of Hong Kong | Collaborative Research Fund (CRF) C6018-19GF | Huating Wang |
| Research Grants Council of Hong Kong | Area of Excellence Scheme (AoE) AoE/M-402/20 | Huating Wang |
| Research Grants Council of Hong Kong | General Research Funds (GRF) 14115319 | Huating Wang |
| Research Grants Council of Hong Kong | General Research Funds (GRF) 14100620 | Huating Wang |
| Research Grants Council of Hong Kong | General Research Funds (GRF) 14106521 | Huating Wang |
| Research Grants Council of Hong Kong | General Research Funds (GRF) 14120420 | Hao Sun |
| Research Grants Council of Hong Kong | General Research Funds (GRF) 14120619 | Hao Sun |
| Research Grants Council of Hong Kong | General Research Funds (GRF) 14103522 | Hao Sun |
| Health and Medical Research Fund | Project Code:08190626 | Huating Wang |

The funders had no role in study design, data collection and interpretation, or the decision to submit the work for publication.

### Author contributions

Yulong Qiao, Conceptualization, Data curation, Formal analysis, Validation, Investigation, Visualization, Methodology, Writing – original draft, Writing – review and editing; Qiang Sun, Data curation, Software, Formal analysis, Investigation, Visualization, Writing – original draft, Writing – review and editing; Xiaona Chen, Investigation, Performed YTHDF1 and YTHDF2 western blot in ASCs and helped with some mouse related experiments; Liangqiang He, Investigation, Generated the C2C12 Ostir expressing line; Di Wang, Investigation, Performed LACE-seq of YTHDC1 in C2C12 and ASCs; Ruibao Su, Investigation, Performed LACE-seq of YTHDC1 in C2C12 and ASCs; Yuanchao Xue, Supervision, Project administration; Hao Sun, Conceptualization, Supervision, Funding acquisition, Writing – original draft, Project administration, Writing – review and editing; Huating Wang, Conceptualization, Resources, Supervision, Funding acquisition, Writing – original draft, Project administration, Writing – review and editing

### Author ORCIDs

Yulong Qiao (iD) http://orcid.org/0000-0002-1952-3077
Huating Wang (iD) http://orcid.org/0000-0001-5474-2905

## Ethics

All animal handling procedures and protocols were approved by the Animal Experimentation Ethics Committee (AEEC) of CUHK (Ref. No. 19-251-MIS). All animal experiments with mice followed the regulations and guidance of laboratory animals set in CUHK.

## Decision letter and Author response

Decision letter https://doi.org/10.7554/eLife.82703.sa1
Author response https://doi.org/10.7554/eLife.82703.sa2

---

# Additional files

## Supplementary files

- Supplementary file 1. Transcriptomic changes upon YTHDC1 inducible knock out (iKO) in ASC.
- Supplementary file 2. LACE-seq profiling of YTHDC1 binding in ASC and C2C12 myoblasts.
- Supplementary file 3. Analysis of YTHDC1 splicing regulation in ASC and C2C12.
- Supplementary file 4. Analysis of YTHDC1 mRNA nuclear export in ASC and C2C12 myoblasts.
- Supplementary file 5. Co-immunoprecipitation/mass spectrometry (co-IP/MS) identifies YTHDC1 interacting proteins in C2C12 myoblasts.
- Supplementary file 6. Sequences of oligos used in the study.
- MDAR checklist

## Data availability

Sequencing data have been deposited in GEO under accession number GSE210127.

The following dataset was generated:

| Author(s) | Year | Dataset title | Dataset URL | Database and Identifier |
|---|---|---|---|---|
| Qiao Y, Sun Q, Chen X, He L, Wang D, Su R, Xue Y, Sun H, Wang H | 2022 | Nuclear m6A Reader YTHDC1 Promotes Muscle Stem Cell Activation/ Proliferation by Regulating mRNA Splicing and Nuclear Export | https://www.ncbi. nlm.nih.gov/geo/ query/acc.cgi?acc= GSE210127 | NCBI Gene Expression Omnibus, GSE210127 |

The following previously published datasets were used:

| Author(s) | Year | Dataset title | Dataset URL | Database and Identifier |
|---|---|---|---|---|
| Gheller BJ, Blum JE | 2020 | A defined N6-methyladenosine (m6A) profile conferred by METTL3 regulates muscle stem cell/myoblast state transitions | https://www.ncbi. nlm.nih.gov/geo/ query/acc.cgi?acc= GSE144885 | NCBI Gene Expression Omnibus, GSE144885 |
| Ding Y | 2021 | CRISPR/Cas9/AAV9-sgRNA Mediated In Vivo Genome Editing Reveals the Indispensability of Myc During Muscle Stem Cells Activation by Remodeling the 3D Chromatin | https://www.ncbi. nlm.nih.gov/geo/ query/acc.cgi?acc= GSE134529 | NCBI Gene Expression Omnibus, GSE134529 |

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
