## [Editor Report]

This valuable study has convincingly identified a specific regulator in skeletal muscle regeneration through a series of elegant experiments. It will form a foundation for further mechanistic investigation. The work will be of future importance in the clinical management of muscle injury and promotion of regeneration.

---

## [Decision Letter]

**Decision letter after peer review:**

Thank you for submitting your article "Nuclear m6A Reader YTHDC1 Promotes Muscle Stem Cell Activation/Proliferation by Regulating mRNA Splicing and Nuclear Export" for consideration by *eLife*. Your article has been reviewed by 3 peer reviewers, and the evaluation has been overseen by a Reviewing Editor and Mone Zaidi as the Senior Editor. The following individual involved in the review of your submission has agreed to reveal their identity: Hong Cheng (Reviewer #1).

Essential revisions:

All three reviewers were positive and supportive but had significant suggestions for revision prior to acceptance.

(A) Reviewer 1 confirmed the view that your paper provides new insight into YTHDC1 function in regulating SC activation/proliferation but added that some of the data could be improved to fully support the conclusions. Specifically:

1. The title "Nuclear m6A Reader YTHDC1 Promotes Muscle Stem Cell Activation/Proliferation by Regulating mRNA Splicing and Nuclear Export" seems a bit overstated. Their data are not sufficient to show YTHDC1 regulating nuclear export. From figure 6 we could see some mRNAs export was inhibited upon YTHDC1 loss but intron retention also occurs on these mRNAs, for example, Dnajc14. Since intron retention could lead to mRNA nuclear retention, the mRNA export inhibition may be caused by splicing deficiency. From the data they provided we could not draw the conclusion that YTHDC1 directly affects mRNA export. I think they could not emphasize this point in the title.

2. The mechanism of YTHDC1 promoting muscle stem cell activation/proliferation is not solidified. The authors could strengthen their evidence through bioinformatics analysis or give more discussion. Besides, the previous work done by Zhao and colleagues (Zhao et al.,. Nature 542, 475-478 (2017).) reported another m6A reader Ythdf2 promotes m6A-dependent maternal mRNA clearance to facilitate zebrafish maternal-to-zygotic transition. Does YTHDC1 regulate mRNA clearance during SC activation/proliferation? The authors should explore this possibility by deep-seq data analysis and give some discussion.

3. The authors should further improve the logical connections between and within paragraphs. For example, the narration of results from lines 138 to 148 would be better if it goes on as a series of phenomenal comparisons between control and iKO muscles but not as bulk descriptions for different groups. More specifically, in lines 139-140 the author described "massive immune cell infiltration was not observed on the first-day post-injury (data not shown)" supposedly of the control group (without any clear indications in this sentence), while in lines 145 and figure 2D immune infiltration was compared between control and iKO muscles. There are also cases where the wording falls short of scientific interpretations such as in line 200, the phrase "since there was no difficulty growing and obtaining C2C12 cells" is quite inappropriate.

4. It is quite confusing to have an AID-YTHDC1 cell line starting in line 178. The authors might want to further explain the purpose of this design. In line 168 the authors used the phrase "early stages" and appeared to focus mainly on the early stage defects in the regeneration of satellite cells. So it is necessary to clarify how C2C12 as a more later-stage myoblast could help explain the mechanism/effect of YTHDC1 loss on SC regeneration as the subtitle described "Inducible YTHDC1 knockout impairs SC activation/proliferation."

5. In figure 2G, the X-axis labels are missing.

6. In figure 3B and 3D, the IF background is too high. The authors need to provide higher-quality images. 3D, the arrows are off-target.

(C) Reviewer (3) was similarly positive, flagging that these preliminary analyses provide a valuable foundation for further mechanistic investigation. The identification of YTHDC1 as a regulator in skeleton muscle development would be beneficial for the field of muscle injury and regeneration.

Reviewer (3) had the following concerns and suggestions:

1. Does skeleton muscle cell activation and proliferation depend on YTHDC1's m6A binding function? Rescue experiments by overexpressing YTHDC1 and its m6A-binding mutants would help.

2. Validation of meRIP and LACE-seq-identified m6A-targets by inactivating m6A writers.

3. What are the roles of YTHDC1's splicing and export targets in skeleton muscle development?

4. hnRNPG is identified as YTHDC1's partner in RNA splicing, what is the role of hnRNPG in skeleton muscle development?

*Reviewer #1 (Recommendations for the authors):*

1. The authors should further improve the logical connections between and within paragraphs. For example, the narration of results from lines 138 to 148 would be better if it goes on as a series of phenomenal comparisons between control and iKO muscles but not as bulk descriptions for different groups. More specifically, in lines 139-140 the author described "massive immune cell infiltration was not observed on the first-day post-injury (data not shown)" supposedly of the control group (without any clear indications in this sentence), while in lines 145 and figure 2D immune infiltration was compared between control and iKO muscles. There are also cases where the wording falls short of scientific interpretations such as in line 200, the phrase "since there was no difficulty growing and obtaining C2C12 cells" is quite inappropriate.

2. It is quite confusing to have an AID-YTHDC1 cell line starting in line 178. The authors might want to further explain the purpose of this design. In line 168 the authors used the phrase "early stages" and appeared to focus mainly on the early-stage defects in the regeneration of satellite cells. So it is necessary to clarify how C2C12 as a more later-stage myoblast could help explain the mechanism/effect of YTHDC1 loss on SC regeneration as the subtitle described "Inducible YTHDC1 knockout impairs SC activation/proliferation."

3. In figure 2G, the X-axis labels are missing.

4. In figure 3B and 3D, the IF background is too high. The authors need to provide higher-quality images. 3D, the arrows are off-target.

*Reviewer #2 (Recommendations for the authors):*

I suggest that the authors should rewrite the manuscript in a more concise manner.

Given the multiple readers in satellite cells, it remains unclear whether YTHDC1 selectively interprets a subset of m6A or all m6A in a non-discrimination manner. In other words, does YTHDC1 play redundant or unique compartment and stage-dependent functions in satellite cells? Another question is if YTHDC1 could also exert an additional function in addition to as a reader of m6A. These should be discussed.

Figure 1B-C is meant to support the dynamics of m6A-related genes, especially Ythdc1. However, the dynamics of Ythdc1 were not obvious from these panels (especially in 1C). The results are also in contrast with the western blot results in Figure 1E. Please clarify.

H-E staining in Figure 2 shows that there was no regeneration whatsoever in the KO, but the laminin staining shows many signals. Are these ghost fibers or other cell types?

Immunofluorescence images in Figure 3 are very difficult to visualize.

*Reviewer #3 (Recommendations for the authors):*

I have the following concerns and suggestions:

1. Does skeleton muscle cell activation and proliferation depend on YTHDC1's m6A binding function? Rescue experiments by overexpressing YTHDC1 and its m6A-binding mutants would help.

2. Validation of meRIP and LACE-seq-identified m6A-targets by inactivating m6A writers.

3. What are the roles of YTHDC1's splicing and export targets in skeleton muscle development?

4. hnRNPG is identified as YTHDC1's partner in RNA splicing, what is the role of hnRNPG in skeleton muscle development?

---

## [Author Response]

Essential revisions:All three reviewers were positive and supportive but had significant suggestions for revision prior to acceptance.(A) Reviewer 1 confirmed the view that your paper provides new insight into YTHDC1 function in regulating SC activation/proliferation but added that some of the data could be improved to fully support the conclusions. Specifically:1. The title "Nuclear m6A Reader YTHDC1 Promotes Muscle Stem Cell Activation/Proliferation by Regulating mRNA Splicing and Nuclear Export" seems a bit overstated. Their data are not sufficient to show YTHDC1 regulating nuclear export. From figure 6 we could see some mRNAs export was inhibited upon YTHDC1 loss but intron retention also occurs on these mRNAs, for example, Dnajc14. Since intron retention could lead to mRNA nuclear retention, the mRNA export inhibition may be caused by splicing deficiency. From the data they provided we could not draw the conclusion that YTHDC1 directly affects mRNA export. I think they could not emphasize this point in the title.

Thanks for the suggestion. It is true that in our initial submission, we had more data to support YTHDC1 regulation of mRNA splicing but not enough on nuclear export. It will take substantial amount of time and efforts to have thorough dissection on both mechanisms. Nevertheless, we argue that our data does provide evidence on YTHDC1 regulation of nuclear export. For example, in Figures 6 C, H, and M, only ~20% of the target mRNAs (such as *Dnaj14*) showed alteration in both splicing and export upon YTHDC1 loss while the majority of the export targets showed no splicing deficiency. For example, *Btbd7* and *Tiparp* in Figure 6 N showed no intron retention. In addition, we have now performed Co-IP experiments to validate the interaction between YTHDC1 and THOC7 (new result added in Figure 7L), which provides extra evidence to support YTHDC1 function in regulating mRNA nuclear export. We thus would like to keep the original title in order to reflect the multifaceted function of YTHDC1 in muscle stem cells.

2. The mechanism of YTHDC1 promoting muscle stem cell activation/proliferation is not solidified. The authors could strengthen their evidence through bioinformatics analysis or give more discussion. Besides, the previous work done by Zhao and colleagues (Zhao et al.,. Nature 542, 475-478 (2017).) reported another m6A reader Ythdf2 promotes m6A-dependent maternal mRNA clearance to facilitate zebrafish maternal-to-zygotic transition. Does YTHDC1 regulate mRNA clearance during SC activation/proliferation? The authors should explore this possibility by deep-seq data analysis and give some discussion.

Thanks for the critical comment. For the first concern, we think YTHDC1 promotes muscle stem cell activation/proliferation through the multi-level gene regulatory capabilities of YTHDC1 on both transcriptional and post-transcriptional processes and the myriads of targets regulated by YTHDC1. In addition, with the newly added data, we believe that YTHDC1’s function is largely dependent on its synergism with hnRNPG (Figure 7 K). We have added the discussion in lines 421-427 of the revised text. For the second question, our data showed that YTHDC1 predominantly localizes in the nucleus of SCs and myoblasts (Figure 1 F and G), thus it may not have a role in regulating mRNA clearance in the cytoplasm like YTHDF2. Nevertheless, there are a few existing reports^1, 2^ suggesting its possible role in mRNA degradation and stability which may arise from its transient shuttling to cytoplasm of cells. We have now added this point in lines 469-472 of the revised text.

3. The authors should further improve the logical connections between and within paragraphs. For example, the narration of results from lines 138 to 148 would be better if it goes on as a series of phenomenal comparisons between control and iKO muscles but not as bulk descriptions for different groups. More specifically, in lines 139-140 the author described "massive immune cell infiltration was not observed on the first-day post-injury (data not shown)" supposedly of the control group (without any clear indications in this sentence), while in lines 145 and figure 2D immune infiltration was compared between control and iKO muscles. There are also cases where the wording falls short of scientific interpretations such as in line 200, the phrase "since there was no difficulty growing and obtaining C2C12 cells" is quite inappropriate.

Thanks for the suggestion. We apologize for the poor logic. We have now edited throughout the text to improve the logical connections between and within paragraphs. And also modified the mentioned sentences. Please see the changes in lines 127-139 and lines 196-198 in the revised text.

4. It is quite confusing to have an AID-YTHDC1 cell line starting in line 178. The authors might want to further explain the purpose of this design. In line 168 the authors used the phrase "early stages" and appeared to focus mainly on the early stage defects in the regeneration of satellite cells. So it is necessary to clarify how C2C12 as a more later-stage myoblast could help explain the mechanism/effect of YTHDC1 loss on SC regeneration as the subtitle described "Inducible YTHDC1 knockout impairs SC activation/proliferation."

Thanks for the suggestion. We used the phrase "early stages" to emphasize the function of YTHDC1 in the activation/proliferation stage comparing the later differentiation stage during muscle regeneration. C2C12 is a proliferating myoblast cell line, which is an acceptable cell line model to study the proliferation of myoblasts. We have now changed the description in lines 156-158 and 168-170 to avoid misunderstanding.

5. In figure 2G, the X-axis labels are missing.

Sorry for the missing labels. We have now included the X-axis labels in Figure 2G.

6. In figure 3B and 3D, the IF background is too high. The authors need to provide higher-quality images. 3D, the arrows are off-target.

Thanks for the suggestion. We apologize for the poor quality of the images. We have now provided images with better quality in Figures 3B and 3D. We have also adjusted the arrows in Figure 3D.

(C) Reviewer (3) was similarly positive, flagging that these preliminary analyses provide a valuable foundation for further mechanistic investigation. The identification of YTHDC1 as a regulator in skeleton muscle development would be beneficial for the field of muscle injury and regeneration.Reviewer (3) had the following concerns and suggestions:1. Does skeleton muscle cell activation and proliferation depend on YTHDC1's m6A binding function? Rescue experiments by overexpressing YTHDC1 and its m6A-binding mutants would help.

Thanks for the great suggestion. We have now performed rescue experiments by overexpressing WT or m6A-binding mutants of YTHDC1 in the AID-YTHDC1 C2C12 cells. and found only the WT but not mutant YTHDC1 could partially rescue the proliferation defect, solidifying our finding that YTHDC1 promoting myoblast proliferation is dependent on its m6A binding function. The data is included in the revised Figure 3 J and Figure 3-—figure supplement 1 I and lines 177-183 of the revised text.

2. Validation of meRIP and LACE-seq-identified m6A-targets by inactivating m6A writers.

Thanks for the great suggestion. We have now used STM2457^10^ to inactivate the m6A writer METTL3 and found it decreased m6A level (by MeRIP-qPCR) and also YTHDC1 binding (by RIP-qPCR) on its targets. The data is included in Figure 4 L and M and in lines 223-225 and 230-236 of the revised text.

3. What are the roles of YTHDC1's splicing and export targets in skeleton muscle development?

Thanks for the question. In Figure 5-—figure supplement 1 D, we performed the GO analysis of m6A-YTHDC1 splicing targets to uncover that YTHDC1 splicing targets were enriched for mRNA processing, Wnt signaling pathway, RNA splicing, etc. In Figure 6 G and L, we also performed GO analysis of YTHDC1 export targets and found these targets were enriched for histone modification, chromatin organization, etc. These YTHDC1 splicing/export targets were important for SC proliferation. For example, canonical Wnt signaling induces satellite-cell proliferation during adult skeletal muscle regeneration^11^. These data suggest that YTHDC1 regulates a wide range of targets important for SC proliferation and muscle development. We have now revised the text in lines 421-427 to clarify the above points.

4. hnRNPG is identified as YTHDC1's partner in RNA splicing, what is the role of hnRNPG in skeleton muscle development?

Thanks for the question. The function of hnRNPG in skeletal muscle stem cells and muscle regeneration/development has barely been studied. To demonstrate its functional relevancy in SCs, we have now knocked down hnRNPG in activated satellite cells and found SC proliferation was reduced. The data suggests that YTHDC1 and hnRNPG function synergistically in promoting SC proliferation. The data is included in Figure 7K and in lines 326-329 of the revised text.

Reference:

1. Shima, H., Matsumoto, M., Ishigami, Y., Ebina, M., Muto, A., Sato, Y., Kumagai, S., Ochiai, K., Suzuki, T. and Igarashi, K. S-Adenosylmethionine Synthesis Is Regulated by Selective N(6)-Adenosine Methylation and mRNA Degradation Involving METTL16 and YTHDC1. Cell Rep 21, 3354-3363 (2017).

2. Zhang, Z., Wang, Q., Zhao, X., Shao, L., Liu, G., Zheng, X., Xie, L., Zhang, Y., Sun, C. and Xu, R. YTHDC1 mitigates ischemic stroke by promoting Akt phosphorylation through destabilizing PTEN mRNA. Cell Death Dis 11, 977 (2020).

3. He, P.C. and He, C. m(6) A RNA methylation: from mechanisms to therapeutic potential. EMBO J 40, e105977 (2021).

4. Widagdo, J., Anggono, V. and Wong, J.J. The multifaceted effects of YTHDC1-mediated nuclear m(6)A recognition. Trends Genet 38, 325-332 (2022).

5. Sheng, Y., Wei, J., Yu, F., Xu, H., Yu, C., Wu, Q., Liu, Y., Li, L., Cui, X.L., Gu, X., Shen, B., Li, W., Huang, Y., Bhaduri-Mcintosh, S., He, C. and Qian, Z. A Critical Role of Nuclear m6A Reader YTHDC1 in Leukemogenesis by Regulating MCM Complex-Mediated DNA Replication. Blood (2021).

6. Cheng, Y., Xie, W., Pickering, B.F., Chu, K.L., Savino, A.M., Yang, X., Luo, H., Nguyen, D.T., Mo, S., Barin, E., Velleca, A., Rohwetter, T.M., Patel, D.J., Jaffrey, S.R. and Kharas, M.G. N(6)-Methyladenosine on mRNA facilitates a phase-separated nuclear body that suppresses myeloid leukemic differentiation. Cancer Cell 39, 958-972 e958 (2021).

7. Chen, C., Liu, W., Guo, J., Liu, Y., Liu, X., Liu, J., Dou, X., Le, R., Huang, Y., Li, C., Yang, L., Kou, X., Zhao, Y., Wu, Y., Chen, J., Wang, H., Shen, B., Gao, Y. and Gao, S. Nuclear m(6)A reader YTHDC1 regulates the scaffold function of LINE1 RNA in mouse ESCs and early embryos. Protein Cell 12, 455-474 (2021).

8. Xiao, W., Adhikari, S., Dahal, U., Chen, Y.S., Hao, Y.J., Sun, B.F., Sun, H.Y., Li, A., Ping, X.L., Lai, W.Y., Wang, X., Ma, H.L., Huang, C.M., Yang, Y., Huang, N., Jiang, G.B., Wang, H.L., Zhou, Q., Wang, X.J., Zhao, Y.L. and Yang, Y.G. Nuclear m(6)A Reader YTHDC1 Regulates mRNA Splicing. Mol Cell 61, 507-519 (2016).

9. Webster, M.T., Manor, U., Lippincott-Schwartz, J. and Fan, C.M. Intravital Imaging Reveals Ghost Fibers as Architectural Units Guiding Myogenic Progenitors during Regeneration. Cell Stem Cell 18, 243-252 (2016).

10. Yankova, E., Blackaby, W., Albertella, M., Rak, J., De Braekeleer, E., Tsagkogeorga, G., Pilka, E.S., Aspris, D., Leggate, D., Hendrick, A.G., Webster, N.A., Andrews, B., Fosbeary, R., Guest, P., Irigoyen, N., Eleftheriou, M., Gozdecka, M., Dias, J.M.L., Bannister, A.J., Vick, B., Jeremias, I., Vassiliou, G.S., Rausch, O., Tzelepis, K. and Kouzarides, T. Small-molecule inhibition of METTL3 as a strategy against myeloid leukaemia. Nature 593, 597-601 (2021).

11. Otto, A., Schmidt, C., Luke, G., Allen, S., Valasek, P., Muntoni, F., Lawrence-Watt, D. and Patel, K. Canonical Wnt signalling induces satellite-cell proliferation during adult skeletal muscle regeneration. J Cell Sci 121, 2939-2950 (2008).

12. Liu, J., Gao, M., He, J., Wu, K., Lin, S., Jin, L., Chen, Y., Liu, H., Shi, J., Wang, X., Chang, L., Lin, Y., Zhao, Y.L., Zhang, X., Zhang, M., Luo, G.Z., Wu, G., Pei, D., Wang, J., Bao, X. and Chen, J. The RNA m(6)A reader YTHDC1 silences retrotransposons and guards ES cell identity. Nature 591, 322-326 (2021).

13. Xu, W., Li, J., He, C., Wen, J., Ma, H., Rong, B., Diao, J., Wang, L., Wang, J., Wu, F., Tan, L., Shi, Y.G., Shi, Y. and Shen, H. METTL3 regulates heterochromatin in mouse embryonic stem cells. Nature 591, 317-321 (2021).

14. Roberson, P.A., Romero, M.A., Osburn, S.C., Mumford, P.W., Vann, C.G., Fox, C.D., McCullough, D.J., Brown, M.D. and Roberts, M.D. Skeletal muscle LINE-1 ORF1 mRNA is higher in older humans but decreases with endurance exercise and is negatively associated with higher physical activity. J Appl Physiol (1985) 127, 895-904 (2019).

15. Mumford, P.W., Romero, M.A., Osburn, S.C., Roberson, P.A., Vann, C.G., Mobley, C.B., Brown, M.D., Kavazis, A.N., Young, K.C. and Roberts, M.D. Skeletal muscle LINE-1 retrotransposon activity is upregulated in older versus younger rats. Am J Physiol Regul Integr Comp Physiol 317, R397-R406 (2019).